# SFSWAP is a negative regulator of OGT intron detention and global pre-mRNA splicing

Ashwin Govindan, Nicholas K Conrad*

Department of Microbiology, University of Texas Southwestern Medical Center, Dallas, United States

## eLife Assessment

This study provides **fundamental** insights into the regulation of a retained intron in the mRNA coding for OGT, a process known to be regulated by the O-GlcNAc cycling system, and highlights the functional role of the splicing regulator SFSWAP. The evidence supporting the claims of the authors is **convincing**; the authors performed an elegant state-of-the-art CRISPR knockout strategy and sophisticated bioinformatic analysis to identify SFSWAP as a negative regulator of alternative splicing. The work will be of interest to researchers in the fields of splicing and glycobiology.

*For correspondence:
nicholas.conrad@
utsouthwestern.edu

Competing interest: The authors declare that no competing interests exist.

**Abstract** O-GlcNAcylation is the reversible post-translational addition of β-*N*-acetylglucosamine to serine and threonine residues of nuclear and cytoplasmic proteins. It plays an important role in several cellular processes through the modification of thousands of protein substrates. O-GlcNAc-ylation in humans is mediated by a single essential enzyme, O-GlcNAc transferase (OGT). OGT, together with the sole O-GlcNAcase OGA, form an intricate feedback loop to maintain O-GlcNAc homeostasis in response to changes in cellular O-GlcNAc using a dynamic mechanism involving nuclear retention of its fourth intron. However, the molecular mechanism of this dynamic regulation remains unclear. Using an O-GlcNAc responsive GFP reporter cell line, we identify SFSWAP, a poorly characterized splicing factor, as a trans-acting factor regulating OGT intron detention. We show that SFSWAP is a global regulator of retained intron splicing and exon skipping that primarily acts as a negative regulator of splicing. In contrast, knockdown of SFSWAP leads to reduced inclusion of a 'decoy exon' present in the OGT retained intron which may mediate its role in OGT intron deten-tion. Global analysis of decoy exon inclusion in SFSWAP and UPF1 double knockdown cells indicate altered patterns of decoy exon usage. Together, these data indicate a role for SFSWAP as a global negative regulator of pre-mRNA splicing and positive regulator of intron retention.

## Introduction

Intron retention (IR) refers to the lack of removal of specific introns in a transcript, resulting in full-length introns in an RNA (*Monteuuis et al., 2019*; *Rekosh and Hammarskjold, 2018*). IR that leads to impaired cytoplasmic export and nuclear retention of the transcript is referred to as intron deten-tion (*Boutz et al., 2015*). Detained introns are a common feature of mammalian transcriptomes, but, unlike other forms of alternate splicing, intron detention controls the levels and timing of production of mature mRNA (*Boutz et al., 2015*; *Braunschweig et al., 2014*; *Yap et al., 2012*). Cells regulate the efficiency of splicing of DIs, and therefore mRNA production, in response to specific environmental cues. DI-containing transcripts may serve as nuclear reservoirs that are spliced in response to cellular environment, or they may be transcriptional dead-ends that are subjected to nuclear degradation

pathways (*Boutz et al., 2015*; *Yap et al., 2012*; *Pendleton et al., 2018*; *Mauger et al., 2016*; *Ninomiya et al., 2011*). Intron detention regulates a wide variety of transcripts in mammals, but the environmental cues and mechanisms that regulate intron detention have been defined for only a few of these RNAs.

The environmental cues that regulate intron detention of the human O-linked β-*N*-acetylglucosamine (O-GlcNAc) transferase (OGT) protein have been defined, but the mechanisms of this regulation are not fully understood (*Park et al., 2017*). OGT encodes the sole enzyme responsible for the post-translational modification O-GlcNAc, where UDP-GlcNAc serves as the cofactor for addition of GlcNAc residues to the hydroxyl groups of serine and threonine residues of thousands of nuclear and cytoplasmic proteins (*Kreppel et al., 1997*; *Lubas et al., 1997*; *Bond and Hanover, 2015*; *Mannino and Hart, 2022*). Conversely, OGA (O-GlcNAcase) is the sole enzyme responsible for removal of O-GlcNAc (*Gao et al., 2001*). Cells maintain O-GlcNAc homeostasis by an intricate feedback mechanism involving intron detention of the OGT transcript (*Park et al., 2017*). Under conditions of high cellular O-GlcNAc (such as treatment with the OGA inhibitor thiamet G [TG]), intron 4 of OGT is detained, resulting in reduction in cytoplasmic OGT mRNA. However, under conditions of low cellular O-GlcNAc (such as glucose deprivation or treatment with the OGT inhibitor OSMI-1), the intron is rapidly excised, resulting in increased levels of cytoplasmic OGT mRNA. Interestingly, the OGA transcript is also regulated by intron detention, but with the inverse relationship to O-GlcNAc levels outcomes as OGT (*Tan et al., 2020*).

We previously identified a cis-acting intronic splicing silencer (ISS) situated within OGT intron 4 that is necessary for IR (*Park et al., 2017*). Subsequent work by other groups showed that the ISS overlaps with an ~150 bp unannotated 'decoy exon' possessing multiple weak 5′ and 3′ splice sites (*Parra et al., 2018*). Deletion of the OGT decoy exon/ISS or even blocking of the decoy 5′ splice sites using morpholino oligonucleotides renders splicing of intron 4 constitutive (*Park et al., 2017*; *Parra et al., 2020*). This suggests the somewhat counterintuitive idea that at least partial assembly of the spliceosome around the decoy is necessary for OGT intron detention even though splicing of the decoy exon does not occur. Decoy exons are not unique to OGT and are present in many other retained intron containing genes, specifically those with longer retained introns (*Parra et al., 2018*; *Pirnie et al., 2017*). The decoy exons mediate IR in these genes as well, but the precise mechanism by which decoy exons regulate IR remains unclear.

In addition to the role of OGT intron detention in buffering changes in cellular O-GlcNAc levels, OGT (and consequently O-GlcNAc) has been proposed to be a 'master regulator' of detained intron splicing (*Tan et al., 2020*). Treatment of cells with the OGT inhibitor OSMI-2 for short time periods (~30 min) induces changes in OGT detention to buffer O-GlcNAc levels. However, longer treatment durations (~2 hr) lead to global changes in detained intron splicing. Differential phosphoproteomics after a short OSMI-2 treatment identified a set of proteins enriched in splicing factors. These 'early responders' could potentially mediate downstream effects of OSMI-2 on splicing and included the putative splicing factor SFSWAP.

SFSWAP (splicing factor, suppressor of white-apricot homolog; SFRS8) encodes the essential human homolog of the *Drosophila* splicing factor SWAP (*Denhez and Lafyatis, 1994*). It is an alternate splicing factor containing an RS domain, but it does not possess a canonical RNA-binding domain (*Denhez and Lafyatis, 1994*). It contains two suppressor of white apricot/Prp21 (SURP) domains which mediate its binding to other proteins containing an SURP interaction domain. The second SURP domain of SFSWAP has been shown to interact with the mammalian branchpoint binding protein SF1 in vitro (*Crisci et al., 2015*). However, the identity of its regulatory targets remains unknown. Its sequence contains a number of known phosphorylation sites (*Tan et al., 2020*) indicating that its function might be regulated by phosphorylation. In addition, it regulates alternate splicing of Tau (*Wang et al., 2004*), CD45 and fibronectin (*Sarkissian et al., 1996*) by inhibiting the inclusion of specific exons.

Here, we identify SFSWAP as a regulator of OGT intron detention using a CRISPR knockout screen with an O-GlcNAc responsive GFP reporter cell line. We show that knockout of SFSWAP leads to enhanced splicing of the OGT detained intron, particularly under high-O-GlcNAc conditions, indicating its role as a negative regulator of OGT splicing. We also show that SFSWAP is a global regulator of detained intron splicing and exon skipping, with enhanced splicing of retained introns and increased inclusion of cassette exons upon SFSWAP knockdown. Our results suggest that SFSWAP

regulates OGT intron detention by modulating the inclusion or recognition of the decoy exon present within the detained intron. Finally, global analysis of decoy exon splicing upon SFSWAP knockdown indicates that SFSWAP may regulate decoy exon splicing globally to mediate its effect on intron detention.

## Results

### A GFP splicing reporter that monitors cellular O-GlcNAc levels

To enable genetic screens for identification of trans-acting factors that regulate OGT intron detention, we constructed a GFP-based splicing reporter that responds to cellular O-GlcNAc levels. The reporter consists of the entire intron 4 of OGT (the detained intron) and corresponding exons (exons 4 and 5) flanked upstream and downstream by the efficiently spliced β-globin intron 2 (and corresponding exons 2 and 3) (*Figure 1a*; *Scarborough et al., 2022*). An eGFP reporter upstream of this assembly is driven by a constitutive CMV promoter and translationally separated from the remaining exons by a T2A element (*Donnelly et al., 2001a*). The T2A element induces ribosomal skipping (*Donnelly et al., 2001b*) producing a GFP polypeptide separate from the protein product of the β-globin-OGT exons.

We integrated this reporter into the AAVS1 safe harbor locus of HCT116 cells by TALEN-mediated recombination and isolated clonal cell lines harboring the reporter (*Sanjana et al., 2012*). Treatment of cells with the OGA inhibitor TG induces high O-GlcNAc levels. This should promote detention of OGT intron 4, nuclear retention of the transcript, and reduced expression of the GFP reporter. On the other hand, treatment with OGT inhibitor OSMI-1 induces a low O-GlcNAc condition, which should promote intron 4 splicing, cytoplasmic export of the mRNA, and increased GFP expression (*Figure 1a*, right). RT-PCR analysis of reporter RNA from clonal cell lines yielded a predominant single band corresponding to the mature reporter mRNA indicating the absence of unexpected splicing events (*Figure 1b*). Moreover, this band increased and decreased upon OSMI-1 and TG treatment, as expected. Further screening of reporter lines by northern blot analysis also confirmed that the GFP reporter is responsive to cellular O-GlcNAc levels. Treatment of the reporter line with 1 µM TG for 6 hr led to reduced levels of the spliced product, while treatment with 10 µM OSMI-1 for the same time led to enhanced splicing increased levels of the reporter mRNA (*Figure 1c*). FACS analysis of the reporter lines treated with the inhibitors verified GFP protein expression, with lower and higher GFP fluorescence after TG and OSMI-1 treatment, respectively (*Figure 1d*). To further characterize reporter activity at various O-GlcNAc levels, we treated reporter clones with conditions known to perturb cellular O-GlcNAc levels. Treatment of the reporter cells with glucosamine or the OGA inhibitor PUGNAc (O-(2-Acetamido-2-deoxy-D-glucopyranosylidene)amino *N*-phenyl carbamate) resulted in reduced GFP protein levels similar to TG, while treatment with the GFAT inhibitor DON (6-diazo-5-oxo-L-norleucine) or glucose deprivation led to increased GFP levels (*Figure 1e*). Finally, we validated the reporter line by knockdown of OGT itself, which will reduce cellular O-GlcNAc levels. As expected, we observed increased levels of reporter mRNA upon OGT knockdown (*Figure 1f*), and this effect masked the effect of subsequent inhibitor treatments. Together, these results demonstrate that our GFP reporter construct reflects the response of endogenous OGT to intracellular O-GlcNAc levels.

### SFSWAP is a negative regulator of OGT intron 4 splicing

To screen for trans-acting factors that regulate OGT IR, we performed whole genome CRISPR knockout screens using the Brunello knockout library (*Doench et al., 2016*) under three treatment conditions – TG, OSMI-1, or untreated. Treatment of reporter lines with TG brings the cells to a 'low GFP' state under which knockout of negative splicing regulatory factors (i.e., those promoting IR) are expected to result in a 'high GFP' state. Treatment with OSMI-1 results in cells that are 'high GFP' and knockout of factors that promote splicing of the retained intron will result in a 'low GFP' state. Finally, untreated cells can be selected for either 'low GFP' or 'high GFP' events. We performed pilot screens at 100× library coverage with a 10-day knockout time period and 24-hr treatment with inhibitors just before selection for either 'high' or 'low GFP' cells by FACS sorting. Subsequent sequencing of the guide RNA locus from the sorted cells and statistical analysis using MAGeCK (*Li et al., 2014*) identified candidates to regulate OGT IR. While we did not obtain any hits with the OSMI-1 and untreated 'low GFP' screens, we successfully obtained hits with both the TG (*Supplementary file 1*) and untreated

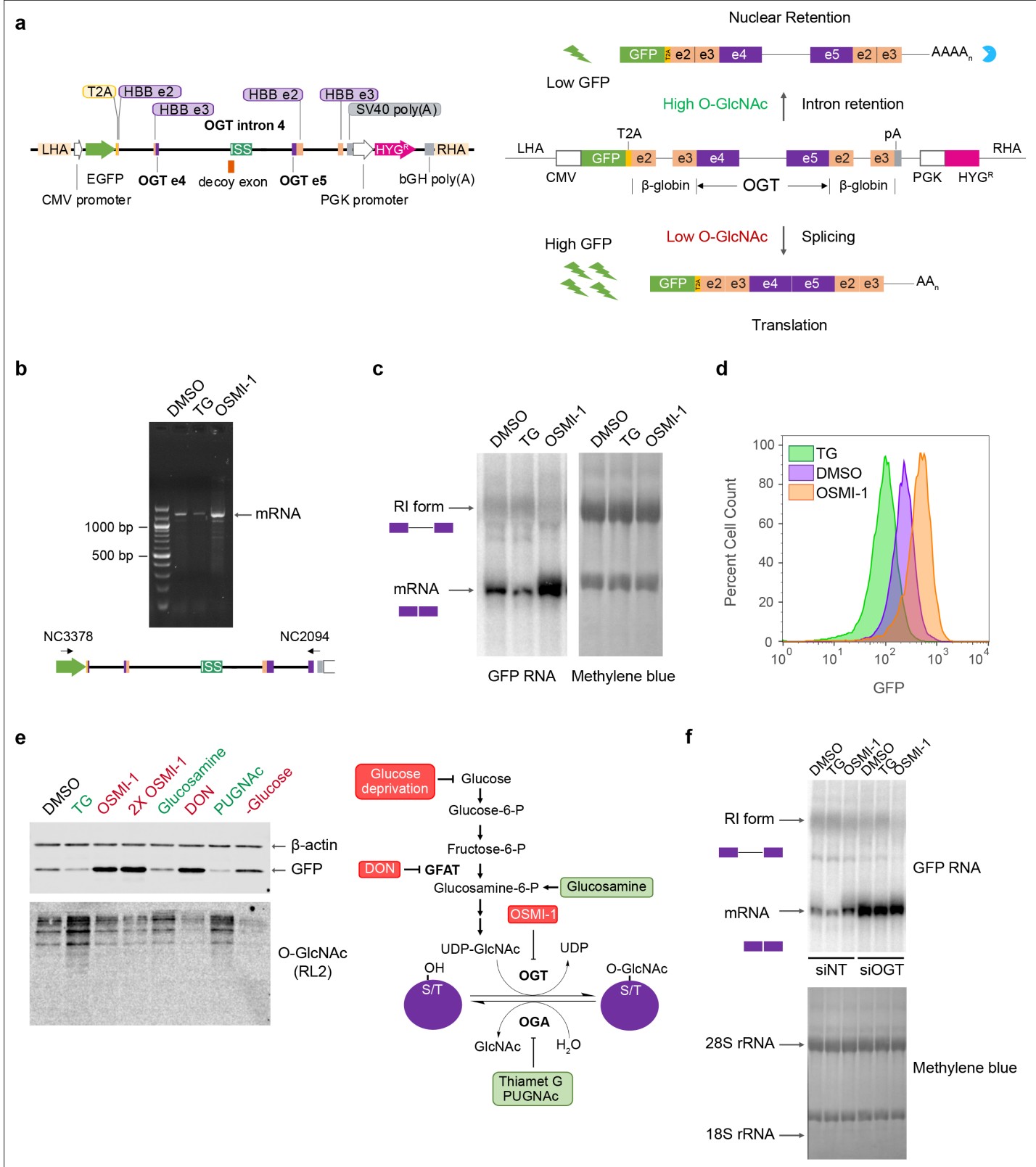

**Figure 1.** Construction of an O-GlcNAc responsive GFP biosensor. (**a**) Schematic of the GFP reporter (left, drawn to scale) and predicted changes in reporter splicing and expression upon varying cellular O-GlcNAc conditions (right). ISS – intronic splicing silencer; LHA – left homology arm; RHA – right homology arm; HBB – hemoglobin subunit β; PGK – phosphoglycerokinase; CMV – cytomegalovirus; bGH – bovine growth hormone; SV40 – Simian virus 40. (**b**) Semi-quantitative RT-PCR of RNA isolated from the reporter line under different treatment conditions using DNA primers (NC3378

*Figure 1 continued on next page*

*Figure 1 continued*

and NC2094) that hybridize within the GFP ORF and just upstream of the polyadenylation signal sequence as shown below. The PCR conditions make it unlikely to detect the full-length detained intron isoform, so only the mRNA is observed. (**c**) Northern blot analysis of total RNA isolated from the reporter line after treatment with either DMSO, 1 µM TG or 10 µM OSMI-1 for 6 hr. The blot was probed for GFP. The retained intron band is heterogeneous and difficult to discern clearly due to its co-migration with the large ribosomal RNA. Methylene blue stain of the blot (right) is shown as a loading control. (**d**) GFP fluorescence levels of the reporter line as measured by flow cytometry after treatment with DMSO, TG, or OSMI-1 for 24 hr. (**e**) Validation of GFP reporter protein levels by western blot analysis after treatment of the reporter line with various modulators of cellular O-GlcNAc levels (left). Steps in the hexosamine biosynthesis pathway targeted by the modulators are shown on the right. Treatment with modulators indicated in red are expected to lead to reduced cellular O-GlcNAc levels, while treatment with those indicated in green are expected to lead to increased cellular O-GlcNAc levels. A broad specificity O-GlcNAc antibody (RL2) and β-actin are used as controls. (**f**) Northern blot analysis of RNA isolated from either 30 nM non-target (siNT) or OGT-specific (siOGT) siRNA-treated reporter line. Cells were treated for 6 hr with DMSO, TG, or OSMI-1 3 days after siRNA treatment. The blot was probed for GFP as above.

The online version of this article includes the following source data for figure 1:

**Source data 1.** Agarose gel of semi-quantitative RT-PCR of RNA isolated from the reporter line under different treatment conditions (source data for *Figure 1b*).

**Source data 2.** Northern blot of total RNA isolated from the reporter line after treatment with either DMSO, 1 µM TG, or 10 µM OSMI-1 for 6 hr (source data for *Figure 1c*).

**Source data 3.** Methylene blue stained loading control for northern blot of RNA isolated from the reporter line under different treatment conditions (source data for *Figure 1c*).

**Source data 4.** Validation of GFP reporter protein levels by western blot analysis after treatment of the reporter line with various modulators of cellular O-GlcNAc levels (source data for *Figure 1e*).

**Source data 5.** Western blot for O-GlcNAc levels (RL2 antibody) after treatment of the reporter line with various modulators of cellular O-GlcNAc levels (source data for *Figure 1e*).

**Source data 6.** Northern blot analysis of RNA isolated from either 30 nM non-target (siNT) or OGT-specific (siOGT) siRNA-treated reporter line (source data for *Figure 1f*).

**Source data 7.** Methylene blue stained loading control for northern blot (source data for *Figure 1f*).

'high GFP' screens (*Supplementary file 2*). Since the list of putative targets from both screens were comparable (*Figure 2—figure supplements 1 and 2*), we performed two additional replicates of the screen at 300× coverage under TG treatment conditions only and obtained similar results.

As expected, the top hit from the screens was OGT itself (*Figure 2a*). In addition, we obtained many hits corresponding to genes involved in glucose metabolism including GFAT (*Figure 1e*) and SLC2A1 (a glucose transporter), further boosting our confidence that the genes identified in the screen are relevant to O-GlcNAc homeostasis. We also obtained a few hits corresponding to RNA-binding proteins (e.g., HNRNPU, HuR) and proteins involved in m6A modification (ZC3H13, KIAA1429). Interestingly, the large majority of the hits were splicing associated factors and/or components of the spliceosome. The latter was surprising because loss of core spliceosome factors would be expected to result in reduced splicing of the reporter (and thus low GFP levels), but we selected for 'high GFP' cells. Nevertheless, these results indicated that the hits from the screen may be associated with a specific spliceosome-mediated mechanism.

We next validated some of the top hits by siRNA mediated knockdown and northern blot analysis or RT-qPCR for the spliced and retained intron junctions of the endogenous OGT transcript (*Figure 2—figure supplement 3*). Consistently, knockdown of SFSWAP increased GFP expression (*Figure 2b*) and enhanced splicing of the reporter RNA (*Figure 2c*, left). To check whether SFSWAP knockdown also results in splicing changes in endogenous OGT, we performed northern blot analysis with a probe corresponding to the 3' UTR of OGT mRNA. Knockdown of SFSWAP in these cells under TG treatment conditions resulted in enhanced splicing of the retained intron compared to the non-target control (*Figure 2c*, right) indicating that the action of SFSWAP was not limited to the reporter. To accurately quantify the changes in splicing in OGT upon SFSWAP knockdown, we performed RT-qPCR on OGT RNA isolated from either non-target or SFSWAP knockdown cells under various conditions (*Figure 2d*). Knockdown of SFSWAP resulted in a significant increase in the spliced form of OGT RNA compared to non-target both under DMSO (~1.8-fold) and TG-treated conditions (~3-fold). In addition, the timing of enhanced intron 4 removal coincided with the timing of decrease in SFSWAP mRNA levels upon knockdown, indicating a direct effect of SFSWAP knockdown on OGT intron 4 splicing (*Figure 2—figure supplement 4*). Knockdown in OSMI-1 treatment conditions did

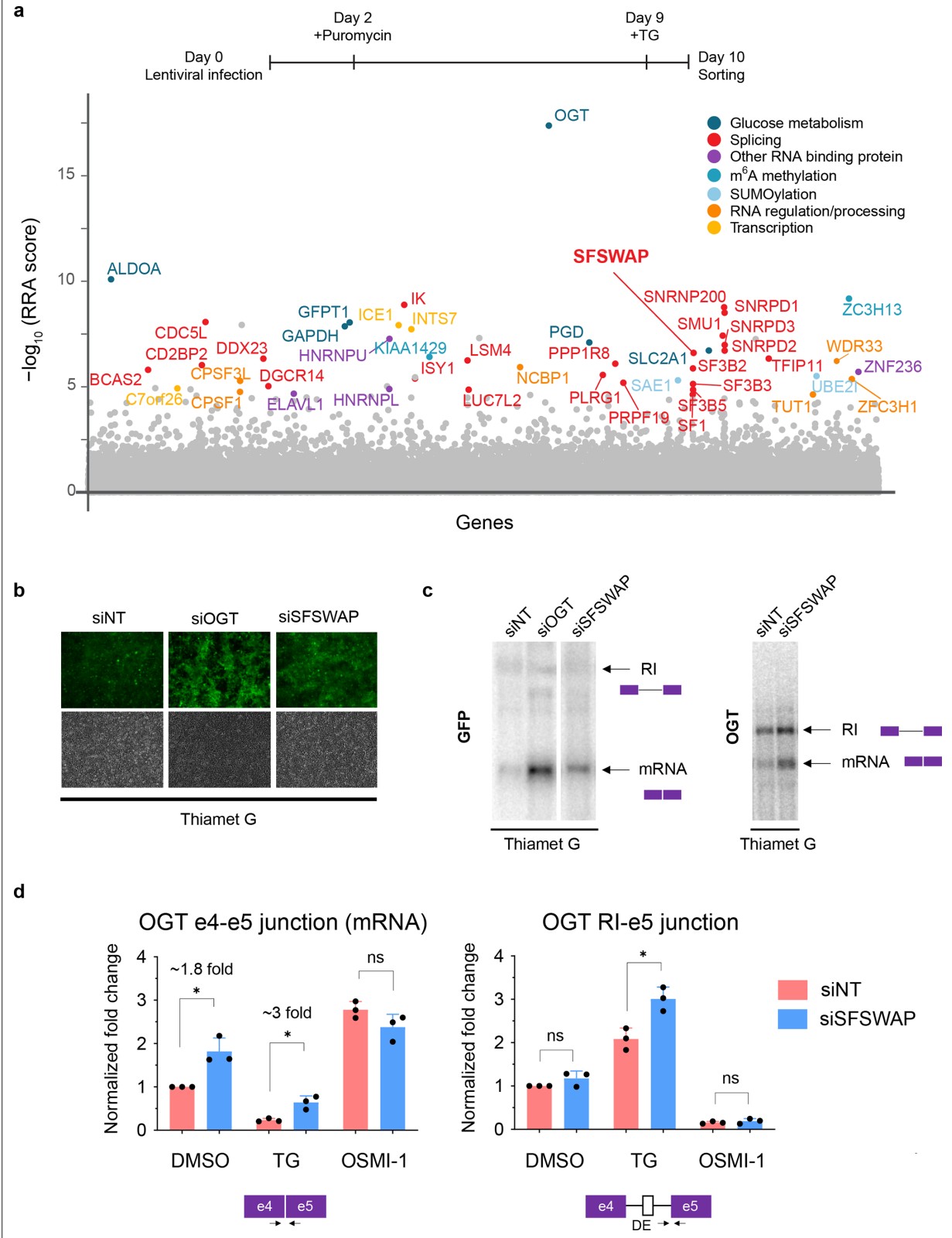

**Figure 2.** SFSWAP is a negative regulator of OGT intron 4 splicing. (**a**) Top, timeline of CRISPR screen. Bottom, MAGeCK analysis of CRISPR screen results from TG-treated gain of GFP screen in three biological replicates. Top hits are color coded based on predicted function of the protein. Target genes are arranged alphabetically on the x-axis. (**b**) GFP fluorescence of TG-treated reporter cells 4 days post-treatment with siRNA corresponding to non-target (siNT), OGT (siOGT), or SFSWAP (siSFSWAP). Bottom panels show corresponding brightfield images. (**c**) Northern blot analysis of RNA

*Figure 2 continued on next page*

**Figure 2 continued**

isolated from either the TG-treated reporter line (left, probed for GFP) or TG-treated 293A-TOA cells (right, probed for OGT) 4 days after treatment with siRNA corresponding to either non-target (siNT) or SFSWAP (siSFSWAP). Cells were treated with TG for 6 hr just before RNA isolation. (**d**) RT-qPCR analysis of the splice junctions of interest after treatment of cells with either DMSO, TG, or OSMI-1 in the presence or absence of SFSWAP knockdown (*n* = 3). Primers used correspond to either the OGT intron 4 spliced junction (e4–e5) or retained intron junction (RI–e5). p-values are derived from unpaired *t*-tests against the corresponding non-target control. Single asterisk (*) denotes p-value <= 0.05.

The online version of this article includes the following source data and figure supplement(s) for figure 2:

**Source data 1.** Northern blot analysis of RNA isolated from TG-treated reporter line and probed for GFP (source data for *Figure 2c*).

**Source data 2.** Northern blot analysis of RNA isolated from 293A-TOA cells and probed for OGT (source data for *Figure 2c*).

**Figure supplement 1.** CRISPR screen for the identification of factors regulating OGT intron detention in the absence of TG treatment (*n* = 1).

**Figure supplement 2.** MAGeCK analysis of TG-treated (top) and untreated (high GFP, bottom) CRISPR screens plotted as a volcano plot.

**Figure supplement 3.** Validation of targets identified in the TG-treated CRISPR screen by RT-qPCR analysis of endogenous OGT intron 4 splicing after knockdown of the target of interest (*n* = 2).

**Figure supplement 4.** Time course RT-qPCR analysis of OGT e4–e5 junction splicing (left) upon SFSWAP knockdown.

not result in a further increase in spliced RNA levels. Together, these data suggest that SFSWAP is a negative regulator of OGT intron 4 splicing.

## SFSWAP is a global regulator of detained intron and skipped exon splicing

Despite its similarity to splicing factors, the importance of SFSWAP in human gene expression and splicing remains unclear. To characterize the functional role of SFSWAP, we performed RNA-seq in either non-target (siNT) or SFSWAP knockdown (siSFSWAP) cells in the absence of TG or OSMI-1 (*Figure 3—figure supplements 1 and 2*). Differential expression analysis using edgeR (*Supplementary file 3*) revealed 252 downregulated and 633 upregulated genes on SFSWAP knockdown, but did not reveal any functional class of RNAs that was regulated by SFSWAP. Since SFSWAP is a predicted splicing factor, we performed alternate splicing analysis using the rMATS package (*Shen et al., 2014*). We analyzed five alternate splicing event types – retained introns (RI), skipped exons (SE), alternate 5′ splice sites (A5SS), alternate 3′ splice sites (A3SS), and mutually exclusive exons (MXE). Interestingly, we found global changes in splicing patterns between siNT and siSFSWAP, with the majority of changes occurring across two of the five event types analyzed – retained introns and skipped exons (*Figure 3a*, *Supplementary file 4*). To visualize this more clearly, we plotted the inclusion level differences between the two conditions (IncLevelDifference, calculated as $\text{IncLevel}_{\text{siNT}} - \text{IncLevel}_{\text{siSFSWAP}}$) against FDR values obtained from rMATS. A positive value for IncLevelDifference indicates more removal of retained introns in SFSWAP knockdown compared to siNT, while a negative value indicates more retention. Knockdown of SFSWAP resulted in increased excision of retained introns globally, indicated by more events with a positive value for IncLevelDiffference (*Figure 3b*) and as exemplified by genome browser shots of IFRD2 and GFUS/TSTA3 (*Figure 3c*). This is consistent with the original screen phenotype, where SFSWAP was isolated as negative regulator of OGT retained intron splicing (*Figure 2*). Previous observations that SFSWAP autoregulates its own expression by control of splicing of its first two retained introns is also consistent with this observation (*Zachar et al., 1987*; *Zachar et al., 1994*). Taken together, these data support the conclusion that SFSWAP promotes IR of a wide variety of transcripts.

In the case of skipped exon events, a positive value for IncLevelDifference indicates increased skipping (reduced inclusion) of the exon upon SFSWAP knockdown, while a negative value indicates increased inclusion. We observed a general enhancement of skipped exon inclusion upon SFSWAP knockdown (*Figure 3b*). This is in agreement with previous studies where SFSWAP was shown to promote exclusion of CD45 exon 4 and the IICS region of fibronectin (*Sarkissian et al., 1996*). Our results are also consistent with the observation that SFSWAP inhibits inclusion of Tau exon 10 (*Wang et al., 2004*). For increased rigor, we also performed alternate splicing analysis using two other pipelines, MAJIQ (*Vaquero-Garcia et al., 2023*) and Whippet (*Sterne-Weiler et al., 2018*; *Figure 3— figure supplements 3 and 4*, *Supplementary file 5*, *Supplementary file 6*). Although the number of differential splicing events identified by these pipelines were lower than with rMATS and largely non-overlapping (*Figure 3—figure supplement 5*), the trends remained similar. These results are

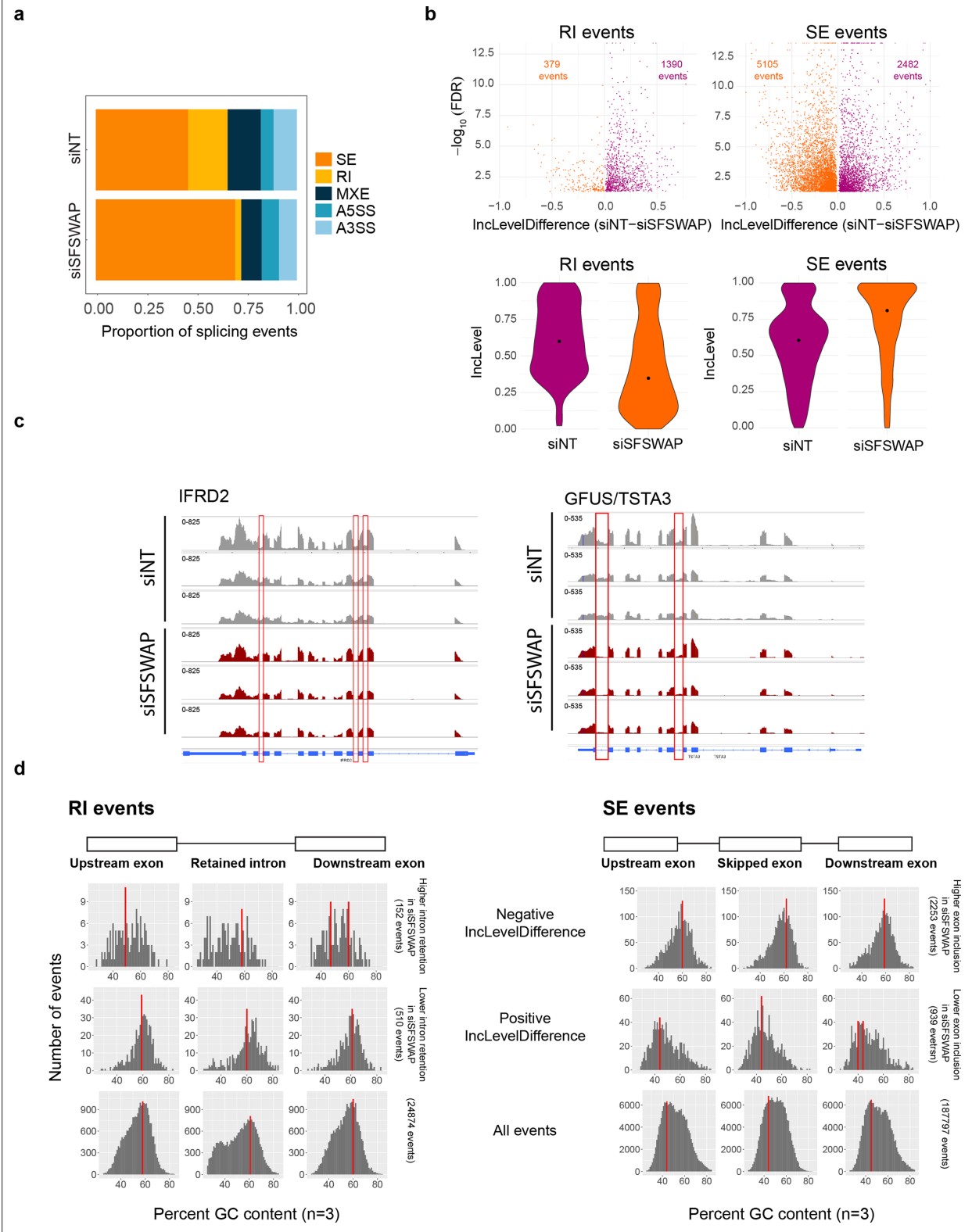

**Figure 3.** SFSWAP is a global regulator of retained intron splicing and exon skipping. (**a**) Alternate splicing analysis in untreated SFSWAP knockdown (siSFSWAP) cells compared to non-target (siNT)-treated cells using rMATS (*n* = 3). The number of events of each type are plotted as proportion of total events detected. A3SS – alternate 3' splice site; A5SS – alternate 5' splice site; MXE – mutually exclusive exon; RI – retained intron; SE – skipped exon. Assignment of sample labels for events was done based on the value of IncLevelDifference (events with positive IncLevelDifference were designated as siNT and negative IncLevelDifference were designated as siSFSWAP). (**b**) Scatter plots of retained intron (RI) and skipped exon (SE) events plotted using

*Figure 3 continued on next page*

*Figure 3 continued*

the JC model of rMATS (top). The difference in inclusion values between siNT and siSFSWAP-treated cells is plotted on the *x*-axis. Only statistically significant events (FDR ≤0.05) are shown. Violin plots of inclusion levels corresponding to the individual samples are shown below. Median inclusion value for the sample is indicated by the black dot. Only significant events with 20% or greater change in inclusion levels are plotted for the violin plots. (**c**) IGV screenshot of read coverages of a few significant retained intron events. Three biological replicates are shown and the intron of interest is marked by the red rectangle. (**d**) GC content of relevant regions of significant RI and SE events (FDR ≤0.05, ≥10% change in inclusion levels). Red bars indicate the mode of GC content in each region.

The online version of this article includes the following source data and figure supplement(s) for figure 3:

**Figure supplement 1.** Validation of SFSWAP knockdown for untreated RNA-seq samples (data shown in *Figure 3*) by western blot analysis (top) and corresponding quantification (normalized to actin, bottom).

**Figure supplement 1—source data 1.** Western blot for validation of SFSWAP knockdown in untreated RNA-seq samples.

**Figure supplement 2.** IGV screenshot showing knockdown of SFSWAP in the untreated RNA-seq samples (data shown in *Figure 3*).

**Figure supplement 3.** Alternate splicing analysis in SFSWAP knockdown background using Whippet (*n* = 3).

**Figure supplement 4.** Alternate splicing analysis in SFSWAP knockdown background using MAJIQ (*n* = 3).

**Figure supplement 5.** Degree of overlap in detected alternately spliced events by rMATS, MAJIQ, and Whippet for retained intron events (left) and skipped exon events (right).

**Figure supplement 6.** IGV screenshot of read coverage of the OGT transcript from RNA-seq analysis in the presence or absence of SFSWAP knockdown (*n* = 3) without any inhibitor treatment.

**Figure supplement 7.** Sashimi plot of splice events around the OGT retained intron showing enhanced removal of the retained intron upon SFSWAP knockdown.

**Figure supplement 8.** Scatter plot of significant RI and SE events from rMATS analysis of RNA-seq data from TG-treated cells in the presence or absence of SFSWAP knockdown.

consistent with a role for SFSWAP in splicing regulation and further suggest a more prominent role as a negative regulator of splicing.

To obtain further insights into the targets of SFSWAP, we looked at the sequence features of these SFSWAP-dependent differentially regulated splice events including length, GC content, and splice site strength. While we did not find any significant differences among the events with respect to length or splice site strength, we observed some interesting trends with respect to GC content. In RIs that are more efficiently spliced upon SFSWAP knockdown (positive IncLevelDifference), we found that the average GC content in the RI and flanking exons was higher compared to those that less efficiently spliced (*Figure 3d*). We observed similar trends toward higher GC content when we examined skipped exon events that result in enhanced inclusion upon SFSWAP knockdown (negative IncLevelDifference). In this case, the skipped exon, upstream and downstream exons all had higher GC content. Together these results suggest that SFSWAP is a global regulator of pre-mRNA splicing that primarily regulates IR and exon skipping events.

## SFSWAP regulates OGT decoy exon inclusion

To delineate the mechanism of SFSWAP regulation of OGT IR, we further looked into our RNA-seq dataset. Interestingly, genome browser visualization revealed no obvious changes in sequence coverage at the OGT retained intron locus after SFSWAP knockdown (*Figure 3—figure supplement 6*). Although surprising, this agrees with the RT-qPCR results on the retained intron junction upon SFSWAP knockdown (*Figure 2d*), where we saw no statistically significant changes in RI junction usage. We reasoned that the lack of junction usage changes could be because the RNA-seq was performed under untreated conditions, while both the CRISPR screen and confirmatory RT-qPCRs were performed under TG-treated conditions (*Figure 2*, *Figure 2—figure supplement 3*). To test whether the effect of SFSWAP knockdown on the OGT retained intron is more appreciable under TG-treated conditions, we repeated the RNA-seq analysis after 6 hr of TG treatment. Visualization of OGT detained intron splicing under these conditions using a sashimi plot provided support for enhanced removal of the detained intron upon SFSWAP knockdown (*Figure 3—figure supplement 7*). Alternate splicing analysis using this dataset reproduced previous results that SFSWAP is a global regulator of pre-mRNA splicing. The effect of SFSWAP on global IR was, if anything, stronger under TG treatment conditions (*Figure 3—figure supplement 8*).

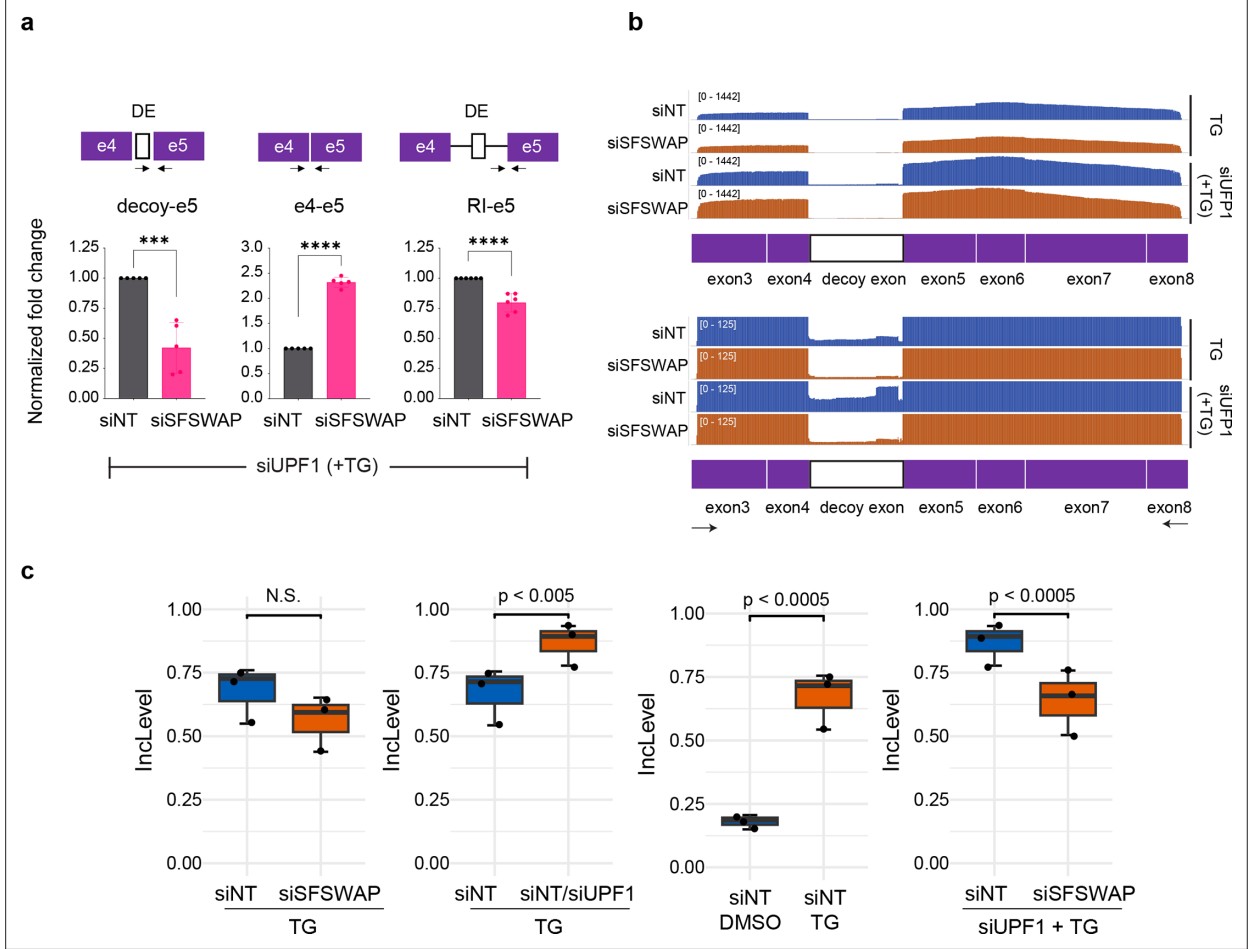

**Figure 4.** SFSWAP regulates OGT decoy exon inclusion. (**a**) RT-qPCR analysis of OGT intron 4 splicing in UPF1 knockdown background. SFSWAP knockdown was performed for 5 days and cells were then treated with TG for 6 hr before RNA isolation and reverse transcription using a mixture of dT$_{20}$ and a DNA oligomer complementary to exon 5 of OGT. qPCR was performed for either the spliced junction (e4–e5), retained intron junction (RI–e5), or decoy–e5 junction as shown. DE – decoy exon. p-values are derived from unpaired *t*-tests against the corresponding non-target controls and are represented as *** (p<0.001) or **** (p<0.0001). (**b**) IGV screenshot of aligned reads after nanopore sequencing of semi-quantitative RT-PCR amplicons generated from the above samples using DNA oligomers complementary to exons 3 and 8 of OGT. A zoomed version is shown below to better show changes in the decoy exon region. (**c**) Quantification of OGT decoy exon inclusion from RNA-seq data in the presence or absence of UPF1 knockdown and/or TG treatment. Inclusion levels and p-values are calculated from the JCEC model of rMATS performed after alignment against a custom reference annotation of the human genome containing decoy exon annotations.

The online version of this article includes the following figure supplement(s) for figure 4:

**Figure supplement 1.** Validation of SFSWAP and UPF1 knockdown in TG-treated RNA-seq samples in the presence or absence of UPF1 knockdown (data shown in *Figures 4 and 5*) by RT-qPCR.

**Figure supplement 2.** IGV screenshot showing knockdown of SFSWAP in the TG-treated RNA-seq samples in the presence (bottom) or absence (top) of UPF1 knockdown (data shown in *Figures 4 and 5*).

Recognition of the decoy exon within the OGT retained intron plays an important part in regulation of IR in OGT (*Park et al., 2017*; *Parra et al., 2018*). Since one of the phenotypes of SFSWAP knockdown is a change in skipped exon splicing, and the decoy mimics a skipped exon internal to the RI, we wanted to test if decoy exon inclusion levels were altered upon SFSWAP knockdown. To do this, we performed RT-qPCR analysis under UPF1 knockdown conditions in the presence of TG (*Figure 4— figure supplements 1 and 2*) to stabilize the decoy exon form, which otherwise would be subject to nonsense-mediated decay (NMD). Under these conditions, we saw reduced inclusion of the decoy exon upon SFSWAP knockdown (*Figure 4a*). We also performed semi-quantitative RT-PCR followed by nanopore sequencing of the region between exons 3 and 8 of OGT mRNA to obtain a comprehensive

picture of splicing in the region. Consistent with the RT-qPCR analysis, there was a ~5-fold reduction in sequencing reads containing the decoy exon upon SFSWAP knockdown (*Figure 4b*).

To further test the role SFSWAP on OGT decoy exon splicing, we analyzed changes in decoy exon inclusion in the TG-treated RNA-seq dataset. The decoy exon in OGT has two predicted 5′ splice sites and two predicted 3′ splice sites. To quantify changes in decoy exon inclusion, we mapped the RNA-seq reads to a version of the human genome where decoy exons resulting from all four possible combinations of splice sites were manually added to the reference annotation. Differential splicing analysis using the JCEC model of rMATS indicated lower inclusion of the decoy exon upon SFSWAP knockdown, as expected, but the differences were not statistically significant (*Figure 4c*, left). As a control, decoy inclusion levels were significantly higher in either in UPF1 knockdown alone (since the decoy exon form is stabilized) or TG treatment alone (*Figure 4c*, middle panels). We reasoned that detection of changes in decoy exon inclusion would be more sensitive in a siSFSWAP/siUPF1 background owing to the presence of the premature stop codon in the decoy exon. Therefore, we performed RNA-seq analysis of SFSWAP knockdown cells (treated with TG) in a UPF1 knockdown background. Alternate splicing analysis using rMATS detected a significant reduction in decoy usage in SFSWAP knockdown cells in this background, consistent with the qPCR results (*Figure 4c*, right). Thus, we conclude that SFSWAP promotes inclusion of the OGT decoy exon. Because decoy exon recognition is a critical component of regulation of OGT intron detention, these data further suggest that SFSWAP regulates OGT intron detention by controlling decoy exon inclusion and/or recognition.

## SFSWAP may be a global regulator of decoy exon usage

Decoy exons have been proposed to regulate IR in a large number of genes in addition to OGT, especially those with longer retained introns (*Parra et al., 2018*). To examine whether SFSWAP regulates decoy exon-mediated IR in genes other than OGT, we re-analyzed our UPF1 knockdown RNA-seq dataset to account for these decoys. Most decoy exons remain uncharacterized to date with the exception of a few like those in OGT (*Park et al., 2017*; *Parra et al., 2018*), ARGLU1 (*Pirnie et al., 2017*), and SF3B1 (*Parra et al., 2018*). To obtain annotations corresponding to putative decoy exons, we used coordinates of novel unannotated cassette exons identified in human erythroblasts by *Parra et al., 2018*. We also obtained a matching list of known retained introns in the same cells identified based on RNA-seq data. We then filtered the list of novel cassette exons to include only those exons whose positions fall entirely within the corresponding retained introns. This list of 2398 novel cassette exons within known retained introns was used as the list of prospective decoy exons (*Supplementary file 7*). In principle, all of these may not be splicing decoys, but we think these are reasonable criteria to assay putative decoys.

We examined the role of SFSWAP in regulating the inclusion of these decoys either in a TG-treated or a TG-treated UPF1 knockdown background by performing alternate splicing analysis using a human reference annotation supplemented with this list of predicted decoy exons. We observed altered patterns of decoy exon inclusion in SFSWAP knockdown conditions compared to non-target. While the changes in retained intron splicing were primarily in the direction of increased splicing (i.e., less RI inclusion) and that of skipped exon splicing were predominantly in the direction of more inclusion upon SFSWAP knockdown, decoy exons were either more or less included (*Figure 5a*). This may reflect the varying mechanisms of action of individual decoys, their splice site strengths, and/or the efficiency of splicing of the intron in which they are found (see Discussion). We further classified these decoy exons based on the predicted translational outcome if the decoys are included in the transcripts (poison cassette, non-poison or untranslated), but we did not see any additional trends for any of these outcome types (*Figure 5b*). We obtained similar trends for decoy exon usage regulation by SFSWAP with our TG-treated dataset in the absence of UPF1 knockdown (*Figure 5—figure supplement 1*), with the only major change being the loss of some of the poison exon events (blue dots) with highest fold changes as expected in the absence of a stabilizing UPF1 knockdown. To further investigate the role of SFSWAP in modulating decoy exon usage, we analyzed the effect of SFSWAP knockdown on splicing of decoy exon-containing retained introns. In contrast to the enhanced removal of retained introns for most non-decoy exon-containing events upon SFSWAP knockdown (*Figure 3b*, *Figure 3—figure supplement 8*), decoy-containing retained intron events did not show a trend toward any direction (*Figure 5c*), suggesting a different mechanism of regulation for decoy-containing retained introns. Consistent with this observation and as observed by other groups (*Boutz et al., 2015*; *Parra*

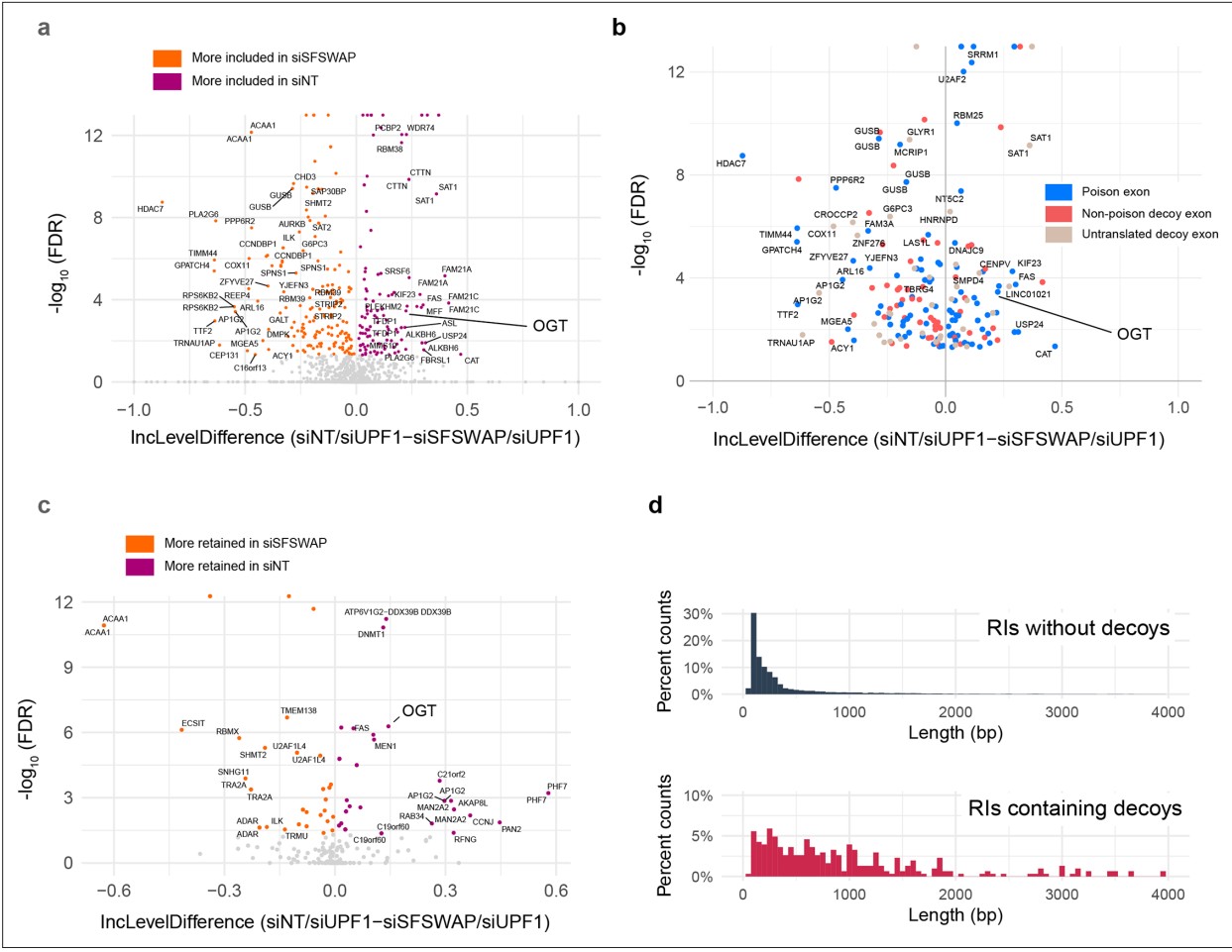

**Figure 5.** SFSWAP is a global regulator of decoy exon splicing. (**a**) Scatter plot of global decoy exon inclusion level changes upon SFSWAP knockdown in a TG-treated UPF1 knockdown background. Statistically significant events are colored. Not all analyzed cassettes may function as splicing decoys. (**b**) Exon types of the events shown in (**a**) classified based on the predicted translation outcome. Events shown in blue introduce an in-frame stop codon in the CDS, thus functioning as poison cassettes. (**c**) Inclusion level changes in decoy-containing retained introns upon SFSWAP knockdown in a TG-treated UPF1 knockdown background. Significant events are colored. (**d**) Length distribution of the decoy-containing retained introns compared to non-decoy-containing retained introns.

The online version of this article includes the following figure supplement(s) for figure 5:

**Figure supplement 1.** SFSWAP is a global regulator of decoy exon splicing.

*et al., 2018*), we observed a trend toward increased intron length for decoy-containing retained introns compared to non-decoy-containing retained introns (*Figure 5d*). Thus, while the mechanism of SFSWAP on individual decoy exons may differ, the data support a role of SFSWAP as a general regulator of decoy exon splicing.

## Discussion

Using a genome-wide CRISPR screen, we identified the putative splicing factor SFSWAP to be necessary for efficient OGT IR. The effect is particularly evident in conditions of high O-GlcNAc that strongly favor retention of intron 4. Additionally, we show that depletion of SFSWAP using siRNAs decreases IR in many transcripts and that skipped exons tend to be more included in these conditions. These observations are wholly consistent with previously published single gene focused studies examining splicing of Tau, fibronectin, CD45, and SFSWAP splicing (*Wang et al., 2004*; *Sarkissian et al., 1996*; *Zachar et al., 1987*). Together, these results strongly support the conclusion that SFSWAP is a splicing factor that primarily functions as a negative regulator of splicing.

In addition, our work suggests that SFSWAP may control OGT splicing by regulating function of its decoy exon and that this may extend to other transcripts with decoy exons. OGT intron 4 and other retained introns require decoy exons for their regulation. Exactly how decoy exons function remains unclear, but several observations suggest that they promote assembly of splicing factors, but not splicing, in order to promote IR (*Park et al., 2017*; *Parra et al., 2018*; *Parra et al., 2020*; *Pirnie et al., 2017*). First, mutation of the weak 5′ splice sites of the OGT decoy exon leads to constitutive splicing of exons 4 and 5 in reporters. Second, eCLIP data demonstrate that U2AF1 and U2AF2 bind to the decoy 3′ splice sites. Third, addition of a morpholino antisense oligonucleotide targeting the OGT decoy's 5′ splice site also promotes constitutive splicing of exons 4–5. Fourth, placing decoy exons and their flanking sequences into other heterologous introns leads to IR, indicating that there is not a gene or intron-specific suppression of splicing. Fifth, CRISPR deletion of the endogenous locus containing the OGT decoy leads to constitutive splicing of exons 4 and 5. Sixth, only a small percentage of mature transcript spliced isoforms splice in the decoy, even when assayed in NMD-inhibiting condition (*Figure 4b*). Together, these data support the recent proposed models that decoy exons require recognition by the spliceosome but are rarely spliced into the final product.

While the mechanism of SFSWAP regulation remains unclear, these characteristics of decoy exons suggest that it functions downstream of exon definition. If SFSWAP suppresses assembly of splicing factors on exons, knockdown of SFSWAP would increase decoy exon definition and thereby increase IR of OGT. In addition, increased recognition of the decoy exon may also increase decoy exon inclusion due to better recruitment of splicing factors assembly on the decoy. Our data show the exact opposite of these predictions as splicing of exons 4–5 increase upon SFSWAP depletion and decoy inclusion decreases (*Figures 2, 4, and 5*).

For these reasons, we favor a model in which SFSWAP functions subsequent to exon definition to repress a downstream event in the splicing cycle (*Figure 6*). For skipped exons and retained introns without decoys, SFSWAP represses splicing at canonical sites which can then be used upon SFSWAP depletion. Perhaps its RS domain acts as a negative regulator by competing with RS domains from canonical SR proteins that promote splicing. For OGT, we speculate that SFSWAP represses inclusion of the decoy similar to how it functions for cassette exons (*Figure 6*, middle and bottom, left). However, in this case, spliceosome assembly at the decoy then indirectly inhibits splicing of exons 4 and 5 by promoting stable unproductive assembly of the spliceosome (*Figure 6*, bottom left). One piece of data that seems to contradict this model is that there is less splicing of the decoy upon SFSWAP depletion (*Figure 4*). We speculate that loss of decoy splicing inhibition by SFSWAP allows competition between the OGT decoy and the exon 4–5 splice sites. Because the latter are stronger sites, decoy inclusion may be lost due to the increase in mRNA production (*Figure 2*). Interestingly, for skipped exons and retained introns more generally, SFSWAP knockdown had a clear tendency to increase splicing efficiency (*Figure 3*). In contrast, SFSWAP knockdown led to a more equal distribution of inclusion and exclusion events on those retained introns containing potential decoy exons (*Figure 5*). Since the putative decoys are generally weaker exons, releasing the negative regulation by SFSWAP may still not be sufficient to promote inclusion of the decoys whereas derepression of canonical skipped exons leads to their inclusion.

Given the absence of canonical RNA-binding domains on SFSWAP, it seems plausible that a second protein with direct RNA-binding activity mediates its activity. Interestingly, we observed enrichment of known SRSF1-binding motifs at the 5′ end of SFSWAP-dependent differentially regulated retained intron events and around differentially regulated exon skipping events (*Figure 6—figure supplement 1*), supporting the idea that SRSF1 acts in concert with SFSWAP to mediate its effect. Alternately, SFSWAP could function in conjunction with the branchpoint binding protein SF1 which has been shown to bind SURP domain containing proteins (like SFSWAP) in vitro (*Crisci et al., 2015*). Notably, we also identified SF1 as a potential target regulating OGT IR in our TG-treated CRISPR screen (*Figure 2*) and could also validate SFSWAP-SF1 interactions in cell lines by co-immunoprecipitation (*Figure 6—figure supplement 2*), further supporting this idea.

With respect to the role of SFSWAP in mediating changes in OGT intron detention in response to varying O-GlcNAc levels in the cell, we speculate that SFSWAP binding to the U2 snRNP is regulated by cellular O-GlcNAc levels through an unidentified sensor. Multiple phosphorylation events on SFSWAP downstream of this sensor may modulate its binding to U2 snRNP. This is supported by the observation that SFSWAP is one of the most highly differentially phosphorylated proteins in the

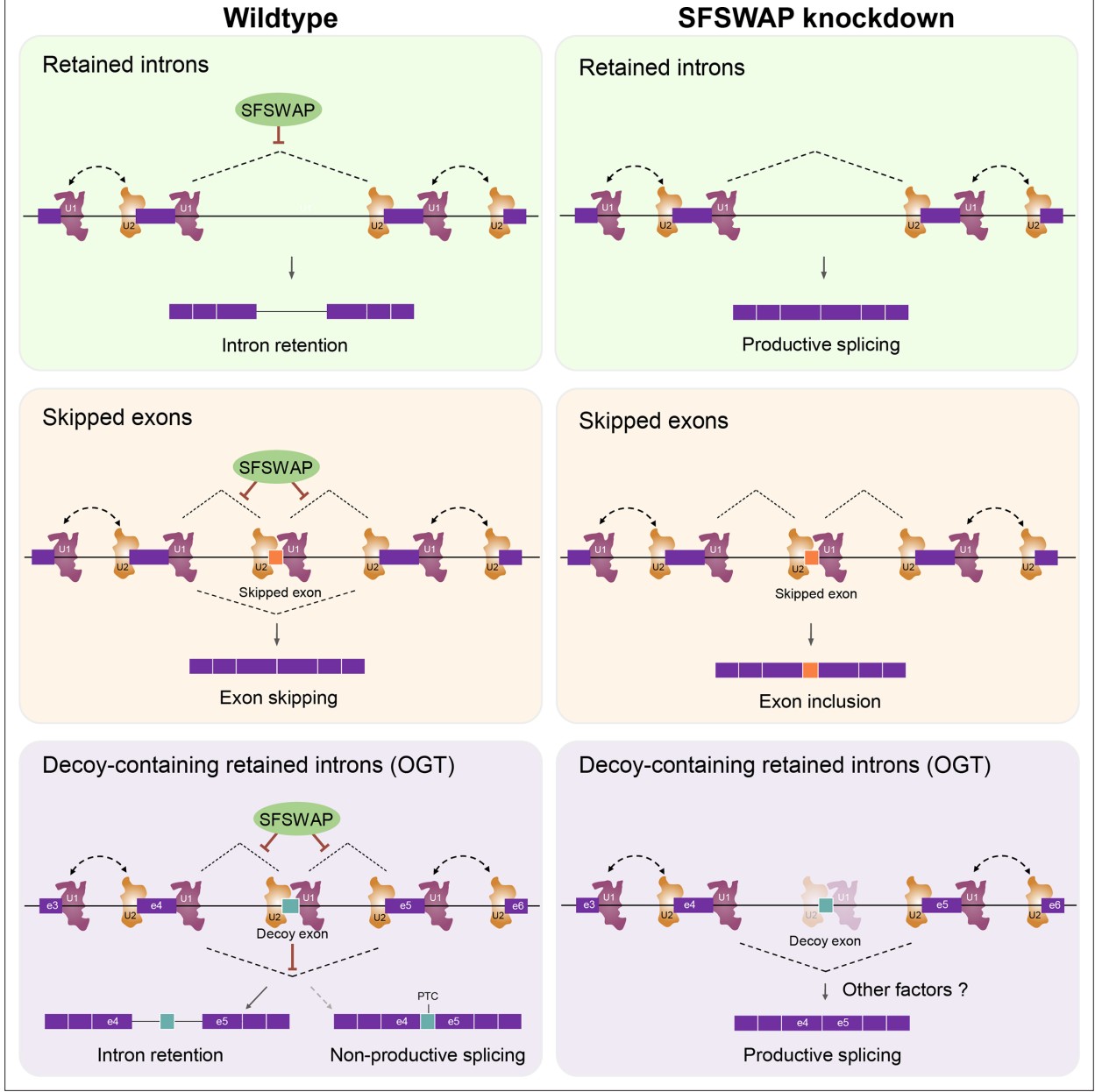

**Figure 6.** Model for the mechanism of action of SFSWAP on intron retention and exon skipping. See text for details. In the case of retained introns without decoy exons or cassette exons (top and middle), we suggest that SFSWAP (green oval) restricts splicing subsequent to definition of the exons by U1 and U2. For retained introns with decoys, this inhibition of decoy exon inclusion supports the decoy exon's function to promote intron retention (see Discussion).

The online version of this article includes the following source data and figure supplement(s) for figure 6:

**Figure supplement 1.** Enrichment of different SRSF1-binding sites on significant alternate splicing events as analyzed by rMAPS.

**Figure supplement 2.** Co-immunoprecipitation (co-IP) analysis of SF1 interaction with SFSWAP.

**Figure supplement 2—source data 1.** Co-immunoprecipitation (co-IP) analysis of SF1 interaction with SFSWAP.

cell after a short treatment (30 min) with an OGT inhibitor, OSMI-2 (*Tan et al., 2020*). Treatment of cells with conditions leading to low O-GlcNAc levels lead to phosphorylation of SFSWAP at multiple residues (primarily S604). We propose that this change in phosphorylation status may regulate its disassociation from U2 snRNP. This then leads to altered recognition of the decoy leading to enhanced removal of the retained intron. On the other hand, high O-GlcNAcylation levels lead to accumulation of an unphosphorylated form of SFSWAP capable of binding to U2 snRNP and mediating decoy exon

recognition, leading to IR. While we show that SFSWAP plays a major role in regulating OGT IR, we also realize that there are likely other factors at play in regulating OGT intron detention, especially given the fact that SFSWAP knockdown does not lead to complete excision of the OGT retained intron as is observed upon deletion of the decoy (*Park et al., 2017*). These factors and their mechanism of action remain a subject for future studies.

## Materials and methods

### Cell culture and growth conditions

HCT116 and HEK293T cells were maintained in DMEM (Sigma, D5796) with penicillin-streptomycin, 2 mM L-glutamine, and 10% fetal bovine serum (FBS, Sigma, F0926) and grown at 37°C in 5% $CO_2$. Media was supplemented with Plasmocin (InvivoGen, ant-mpt, 1:10,000) and 50 µg/ml hygromycin (Sigma H3274) as required. 250 µg/ml hygromycin or 1 µg/ml of puromycin (Sigma P8833) was used for selection of cells. 293A-TOA cells (*Sahin et al., 2010*) were grown similarly, but with Tet-free FBS (Atlanta Biologicals, S10350). Plasmid transfections were performed using TransIT-293 (Mirus bio, MIR 2704) for HEK293 cells or Fugene HD (Promega, E2311) for HCT116 cells according to manufacturer's protocols. For modulation of O-GlcNAc levels, cells were treated with 10 µM OSMI-1 (Sigma SML1621), 1 µM TG (Sigma SML0244), 100 µM DON, 10 mM glucosamine, or 50 µM PUGNAc for 6 hr. Glucose deprivation was performed by growing cells in glucose-free media (Thermo Fisher Scientific, 11966025) for 24 hr. A list of key resources and primer sequences is provided in Appendix 1—key resources table and *Supplementary file 8* respectively.

### O-GlcNAc responsive GFP reporter construction

An 864-bp PCR product encoding the eGFP, the T2A element and a part of β-globin exon 2 was PCR amplified from pNC1330 using the oligomers NC3739 and NC3482 (Acc65I and BamHI ends). A second PCR product (1056 bp) encoding the remaining part of β-globin exon 2, intron 2, and exon 3 was PCR amplified from pNC980 using NC2085 and NC3851 (BamHI and XbaI ends). The two fragments were ligated into an Acc65I XbaI digested pcDNA3 vector to obtain a BamHI site at the junction of the two fragments. A 4008-bp BamHI fragment encoding OGT exon 4, intron 4, and exon 5 from pNC980 was then ligated into the newly generated BamHI site to obtain a pcDNA version of the GFP reporter. The entire reporter region consisting of eGFP, T2A, OGT exon 4, intron 4, exon 5, and flanking β-globin introns and exons was then moved to pNC1049 using the Acc65I XbaI sites to obtain the final reporter construct (pNC1771).

### AAVS1 integration and clonal cell line generation

The reporter construct generated above was integrated into the AAVS1 safe harbor locus of HCT116 cells by TALEN-mediated recombination as described before (*Scarborough et al., 2021*). hAAVS1 1L TALEN and hAAVS1 1R TALEN were gifts from Feng Zhang (Addgene plasmid #35431 and #35432, respectively) (*Sanjana et al., 2012*). Reporter integrated cells were selected by growth in 250 µg/ml hygromycin for 2 weeks. Fluorescence activated cell sorting was performed to select cells expressing varying GFP levels, which were expanded and screened for reporter activity in the presence of either OSMI-1 or TG by northern blot analysis. Cell lines showing enhanced intron 4 splicing in response to OSMI-1 and increased retention of the reporter RNA in response to TG were selected for further screening by RT-qPCR.

### Reporter validation

For RT-qPCR validation, total RNA isolated from inhibitor treated cells was subjected to reverse transcription using 2.5 µM $dT_{20}$ oligomers and 200 U of SuperScript II reverse transcriptase (Thermo Fisher Scientific, 18064014). The resulting cDNA was used as a template for PCR amplification using NC3378 (binding within the GFP ORF) and NC2094 (binding downstream of the last of β-globin exon 3). An amplicon size of 1251 bp is expected in the absence of any unexpected alternate splicing events. For validation of O-GlcNAc responsiveness, the reporter line was treated with modulators of O-GlcNAc levels as above. Cells were subjected to either FACS analysis or western blot analysis using either anti-GFP antibodies or O-GlcNAc RL2 antibodies. β-Actin antibodies were used for loading controls.

## Northern blot analysis

Northern blot analysis was performed using standard techniques. Briefly, about 3–5 µg of total RNA was resolved on a 0.8–1.4% formaldehyde agarose gel, transferred to a positively changed nylon membrane by capillary transfer, UV-crosslinked and probed with radiolabeled RNA probes generated by in vitro transcription with T7 RNA polymerase. Sequences of DNA oligomers used to generate transcript specific probes are listed in *Supplementary file 8*.

## siRNA knockdown

For SFSWAP knockdowns, 293A-TOA were transfected with 40 nM siRNA using RNAiMAX (Thermo Fisher Scientific, 13778150) following the manufacturer's protocol. Cells were split 24 hr post-transfection and allowed to grow for an additional 4 days. All other gene knockdowns were performed similarly with 30 nM siRNA and 3 additional days of growth after splitting.

## CRISPR screens

Unbiased pooled CRISPR screens using the Human Brunello CRISPR knockout pooled library (a gift from David Root and John Doench, Addgene #73179) (*Doench et al., 2016*) were performed as before (*Scarborough et al., 2021*). Briefly, amplified library DNA isolated from ElectroMAX Stbl4 Competent Cells (Thermo Fisher Scientific, 11635018) was used to generate a high-titer lentiviral library using HEK293T cells. Following viral titer estimation by CellTiter-Glo Luminescent Cell Viability Assay (Promega, G7570) and Benzonase (Sigma, E1014) treatment, the lentiviral supernatant was used to infect reporter cells at a 30% infection ratio. Pilot screens were performed at 100× library coverage and replicate screens were performed at 300× coverage. Infected cells were selected in puromycin for a total of 8 days. On the last day of selection, cells were either left untreated or treated with either TG (replenished every 12 hr for a total of 24 hr) or OSMI-1 for 24 hr. In the case of TG-treated screens, the high GFP-expressing population of cells were collected, while in case of the OSMI-1-treated screens, low GFP-expressing cells were collected. In the case of untreated screens, both populations of cells were collected. Sorting was performed at the Flow Cytometry Core at UT Southwestern Medical Center. Following DNA isolation and PCR amplification of the guide RNA-encoding locus, next-generation sequencing was performed on an Illumina NextSeq550 instrument using an indexed single-end sequencing protocol with a read length of 75 bp. The original DNA preparation used to generate the lentiviral library was sequenced simultaneously to ensure complete library representation. Reads were mapped to the Brunello library guide RNA sequences and statistical analysis to calculate enrichment over unselected cells was performed using the MAGeCK-VISPR (*Li et al., 2015*) pipeline.

## Validation of CRISPR screen hits

Genes corresponding to the top enriched guide RNAs from the TG-treated CRIPSR screen were validated as regulators of OGT intron detention by siRNA-mediated knockdown of the gene in the presence of TG followed by RT-qPCR of relevant junctions of endogenous OGT. Target genes were further validated by northern blot analysis of either the reporter RNA or endogenous OGT RNA following knockdown.

## RT-qPCR analysis

For RT-qPCR analysis of endogenous OGT splicing, 1 µg of total RNA was used for reverse transcription using 2.5 µM $dT_{20}$ and 200 U of M-MuLV Reverse Transcriptase (NEB, M0253S) for 50 min at 42°C. Following heat denaturation at 80°C for 10 min, excess RNA and RNA:DNA hybrids were removed by digestion with RNase A and RNase H. The resulting cDNA was used as a template for qPCR using the iTaq Universal SYBR Green Supermix (Bio-Rad, 1725121). PCR was performed for 40 cycles with an annealing temperature of 60°C. Analysis was performed using the ΔΔCt method using actin for normalization. OGT decoy exon qPCRs were performed similarly in a UPF1 knockdown background, but with 5 µg of total RNA, 200 U of SuperScript IV (Thermo Fisher Scientific, 18090050) and a mixture of 2.5 µM $dT_{20}$ and 2 pmol of an OGT exon 5 specific reverse primer (NC985).

## RNA-seq analysis

RNA-seq analysis was performed in 293A-TOA cells in either wild-type or UPF1 knockdown background with or without 1 μM TG treatment for 6 hr. All RNA-seq experiments were performed in biological triplicates. Cells were harvested from 6-well plates at 70–80% confluency. Knockdown efficiency of proteins of interest were validated by either western blot analysis or RT-qPCR with appropriate primers. RNA isolation was performed using Zymo Direct-zol RNA miniprep kit (Zymo R2050). One μg of total RNA was used for polyA-RNA isolation and library preparation using the KAPA mRNA Hyperprep kit (Roche, KK8580) following the manufacturer's protocol. RNA was fragmented to an average fragment size of 100 bp. The kit-supplied adaptor was replaced with NEB adaptors and index primers (NEB, E7335S) to enable pooling of multiple samples per flow cell. Single-end sequencing with a read length of 100 bp was performed using an Illumina NextSeq 2000 instrument at the McDermott Center Next Generation Sequencing (NGS) Core at UT Southwestern Medical Center to obtain 60–65 million reads per sample on average.

Reads were trimmed with cutadapt and aligned to GENCODE release 40 of the human reference genome with STAR (version 2.7) (*Dobin et al., 2013*) using the '--twopassMode Basic' option. Batch correction was performed, if necessary, using the EDASeq (*Risso et al., 2011*) and RUVSeq (*Risso et al., 2014*) R packages. Differential expression analysis was performed using edgeR (*Robinson et al., 2010*) using factors of unwanted variation obtained from RUVSeq as an additional covariate in the design matrix.

## Alternate splicing analysis

Alternate splicing analysis was performed using rMATS (*Shen et al., 2014*), MAJIQ (*Vaquero-Garcia et al., 2023*), and Whippet (*Sterne-Weiler et al., 2018*). rMATS analysis was performed with either untreated or TG-treated samples using all three biological replicates of siNT and siSFSWAP samples. For rMATS, events with FDR ≤0.05 and IncLevelDifference ≥0.2 were considered significant. Whippet analysis was performed with default parameters using all three biological replicates of untreated siNT or siSFSWAP samples. Events with probability ≥0.9 and ΔPSI of 20% or greater were considered significant. MAJIQ analysis was performed using the GFF3 version of the genome obtained from GENCODE (instead of GTF). The 'build' step was performed with the options `--min-experiments 2 and --simplify` 0.10. Following the 'deltapsi' step, Local Splicing Variations (LSVs) were converted into rMATS-style binary events using the 'modulize' function. Events with probability ≥0.9 and ΔPSI of ≥10% were considered significant. The proportion of event types as a fraction of all event types was plotted from rMATS output (JC model) using MASER (https://github.com/DiogoVeiga/maser; *DiogoVeiga, 2022*). GC content calculations were performed based on rMATS JC model outputs taking only significant events (as defined above) into consideration. Enrichment of RBP-binding sites around alternately spliced events was visualized using the rMAPS2 web server (*Hwang et al., 2020*) with rMATS JCEC model output files provided as inputs. Sashimi plots for regions of interest were generated using rmats2sashimiplot (https://github.com/Xinglab/rmats2sashimiplot; *Xinglab, 2025*).

## Amplicon sequencing

cDNA preparations from TG-treated SFSWAP knockdown (or non-target) cells in a UPF1 knockdown background were used as templates for PCR amplification of the region between OGT exons 3 and 8 using the primers NC1769 and NC988. PCR was performed for 30 cycles using Q5 DNA polymerase (NEB, M0491S). The amplicons were sequenced at Plasmidsaurus. Sequencing reads were aligned to a version of OGT cDNA sequence containing the longest form of the OGT decoy exon (153 bp) but excluding the rest of the retained intron using minimap2 (*Li, 2018*) and the number of reads overlapping the decoy exon were computed using bedtools (*Quinlan and Hall, 2010*). Coordinates for the four possible OGT decoy exon forms are as follows: decoy 143 (chrX:71,546,347–71,546,489), decoy 146 (chrX:71,546,344–71,546,489), decoy 150 (chrX:71,546,347–71,546,496), and decoy 153 (chrX:71,546,344–71,546,496).

## Global alternate decoy exon usage analysis

Decoy exon usage analysis was performed using RNA-seq reads obtained with or without UPF1 knockdown in the presence of TG, and either non-target (siNT) or SFSWAP (siSFSWAP) knockdown. To obtain decoy exon annotations, we used coordinates of novel unannotated cassette exons identified

in human erythroblasts by *Parra et al., 2018* based on a set of stringent criteria. This list of novel cassette exons was further filtered to only keep cassettes located within known retained introns in the same cell line. The coordinates of these novel cassette exons located within known retained introns were used to supplement RefSeq (based on genome assembly GCF_000001405.34/GRCh38.p8) to obtain a version of the human reference annotation containing potential decoy exons. RNA-seq reads were trimmed and mapped to this modified reference using STAR in 'two-pass Basic' mode. rMATS analysis was performed as above.

## Co-immunoprecipitation analysis

Co-immunoprecipitation of SFSWAP with SF1 was performed from 293A-TOA cells grown in a 60-mm plate treated with either DMSO, TG, or OSMI-1. Briefly, cells were lysed in a buffer containing 100 mM NaCl, 2.5 mM $MgCl_2$, 10 mM Tris-HCl pH 7.5 and 0.5% IGEPAL CA-630, 1 mM PMSF and 1× Protease Inhibitor Cocktail Set V (Millipore, 539137). Samples were nutated at room temperature with 20 U RQ1 DNase (Promega, M6101) and RNase A (10 μg/ml) for 15 min, clarified by centrifugation, and incubated with 2 μg of mouse anti-SF1 antibodies for 2 hr at 4°C. The antibody complexes were pulled down with 20 μl protein A magnetic beads, washed five times with the same buffer lacking protease inhibitors and eluted with 1× SDS–PAGE loading buffer. The eluate was resolved on a 7.5% SDS–PAGE gel, transferred to a nylon membrane, probed with rabbit anti-SFSWAP antibodies (1:1000) and detected using IRDye-conjugated secondary antibodies on a Licor instrument.

## Material availability

All unique plasmids and cell lines generated for this study are available upon request to corresponding author.

## Acknowledgements

We thank Drs. Didier Trono, Feng Zhang, David Root, and John Doench for plasmids. We also thank the UT Southwestern Flow Cytometry Core, Proteomics Core, and the McDermott Center Next Generation Sequencing Core for assistance with experiments. This research was supported by the National Institutes of Health R01 GM127311, U01 CA242115, R01 AI153175, and R01 AI123165.

## Additional information

### Funding

| Funder | Grant reference number | Author |
|---|---|---|
| National Institutes of Health | R01 GM127311 | Nicholas K Conrad |
| National Institutes of Health | U01 CA242115 | Nicholas K Conrad |
| National Institutes of Health | R01 AI153175 | Nicholas K Conrad |
| National Institutes of Health | R01 AI123165 | Nicholas K Conrad |

The funders had no role in study design, data collection, and interpretation, or the decision to submit the work for publication.

### Author contributions

Ashwin Govindan, Conceptualization, Data curation, Formal analysis, Methodology, Writing - original draft, Writing - review and editing; Nicholas K Conrad, Conceptualization, Data curation, Supervision, Funding acquisition, Methodology, Writing - original draft, Project administration, Writing - review and editing

### Author ORCIDs

Ashwin Govindan (iD) https://orcid.org/0000-0003-1759-707X

Nicholas K Conrad https://orcid.org/0000-0002-8562-0895

Reviewer #1 (Public review): https://doi.org/10.7554/eLife.104439.3.sa1
Reviewer #2 (Public review): https://doi.org/10.7554/eLife.104439.3.sa2
Reviewer #3 (Public review): https://doi.org/10.7554/eLife.104439.3.sa3
Author response https://doi.org/10.7554/eLife.104439.3.sa4

## Additional files

### Supplementary files

Supplementary file 1. Full list of putative target genes identified by MAGeCK analysis of TG-treated gain of GFP CRISPR screen (*n* = 3).

Supplementary file 2. Full list of putative target genes identified by MAGeCK analysis of untreated gain of GFP CRISPR screen (*n* = 1).

Supplementary file 3. List of differentially expressed genes upon SFSWAP knockdown (unreated samples) as determined by edgeR analysis.

Supplementary file 4. List of differential splicing events upon SFSWAP knockdown (untreated samples) as analyzed by rMATS.

Supplementary file 5. List of differential splicing events upon SFSWAP knockdown (untreated samples) as analyzed by Whippet.

Supplementary file 6. List of differential splicing events upon SFSWAP knockdown (untreated samples) as analyzed by MAJIQ.

Supplementary file 7. List of potential decoy exons analyzed in *Figure 5*.

Supplementary file 8. List of primers used.

MDAR checklist

### Data availability

Raw sequencing reads for all sequencing experiments described in this paper have been deposited in NCBI GEO with the accession number GSE277952. Full-length gels are included for all figures in a separate source file.

The following dataset was generated:

| Author(s) | Year | Dataset title | Dataset URL | Database and Identifier |
|---|---|---|---|---|
| Govindan A, Conrad NK | 2024 | SFSWAP is a negative regulator of OGT intron detention and global pre-mRNA splicing | https://www.ncbi.nlm.nih.gov/geo/query/acc.cgi?acc=GSE277952 | NCBI Gene Expression Omnibus, GSE277952 |

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

# Appendix 1

**Appendix 1—key resources table**

| Reagent type (species) or resource | Designation | Source or reference | Identifiers | Additional information |
|---|---|---|---|---|
| Cell line (*H. sapiens*) | HEK293A-TOA | Dr. Nicholas K. Conrad; *Sahin et al., 2010* | UT Southwestern Medical Center | |
| Cell line (*H. sapiens*) | HEK293T | Dr. Joshua Mendell | UT Southwestern Medical Center | |
| Cell line (*H. sapiens*) | HCT116 | ATCC | CCL-247 | |
| Cell line (*H. sapiens*) | HCT116 GFP-β-OGT | This paper | Clone E9 | Maintained by Nicholas K. Conrad lab |
| Strain, strain background (*E. coli*) | ElectroMAX Stbl4 competent cells | ThermoFisher | Cat #11635018 | Competent cells |
| Strain, strain background (*E. coli*) | DH5α | ThermoFisher | Cat #EC0112 | Competent cells |
| Antibody | Rabbit polyclonal anti-GFP | Abcam | Cat #ab6556;RRID:AB_305564 | 1:2000 |
| Antibody | Rabbit polyclonal anti-SFSWAP | Bethyl | Cat #A300-985A;RRID:AB_2185354 | 1:1000 |
| Antibody | Rabbit polyclonal anti-SF1 | Abnova | Cat #H00007536-M01;RRID:AB_607016 | 1:5000 |
| Antibody | Goat anti-rabbit IRDye 800CW | LI-COR Biosciences | Cat #926–32211;RRID:AB_621843 | 1:10,000 |
| Antibody | Goat anti-mouse IRDye 800CW | LI-COR Biosciences | Cat #926–32210;RRID:AB_621842 | 1:10,000 |
| Antibody | Mouse monoclonal anti-beta-actin | Abcam | Cat #ab6276; RRID:AB_2223210 | 1:10,000 |
| Antibody | Mouse monoclonal anti-O-GlcNAc (RL2) | Invitrogen | Cat #MA1-072; RRID:AB_326364 | 1:1000 |
| Antibody-based Reagent | Myc-Trap magnetic agarose beads | Chromotek | Cat #ytma | |
| Synthetic Peptide | 2 x Myc-peptide | Chromotek | Cat #2yp | |
| Recombinant DNA reagent | Plasmid: pcDNA3 | ThermoFisher | Cat #V79020 | |
| Recombinant DNA reagent | Plasmid: psPAX2 | Addgene | Plasmid #12260 | psPAX2 was a gift from Didier Trono (Addgene plasmid #12260; http://n2t.net/addgene:12260; RRID:Addgene_12260) |
| Recombinant DNA reagent | Plasmid: pMD2.G | Addgene | Plasmid #12259 | pMD2.G was a gift from Didier Trono (Addgene plasmid #12259; http://n2t.net/addgene:12259; RRID:Addgene_12259) |

*Appendix 1 Continued on next page*

*Appendix 1 Continued*

| Reagent type (species) or resource | Designation | Source or reference | Identifiers | Additional information |
|---|---|---|---|---|
| Recombinant DNA reagent | Plasmid: lentiCRISPR v2 | *Sanjana et al., 2014* | Plasmid #52961 | lentiCRISPR v2 was a gift from Feng Zhang (Addgene plasmid #52961; http://n2t.net/addgene: 52961; RRID:Addgene_52961) |
| Recombinant DNA reagent | Brunello pooled library in lentiCRISPR v2 | *Doench et al., 2016* | Pooled Library #73179 | Human Brunello CRISPR knockout pooled library was a gift from David Root and John Doench (Addgene #73179) |
| Recombinant DNA reagent | Plasmid: pc-Myc-SFSWAP | GenScript | Cat #OHu108377 | Obtained as an N-terminal Myc-tagged clone |
| Recombinant DNA reagent | Plasmid: hAAVS1-GFP-T2A-b2-MAT-E8-3 hp2-6m9 | *Scarborough et al., 2021* | pNC1330 | |
| Recombinant DNA reagent | Plasmid: hAAVS1-GFP-β-OGT | This paper | pNC1771 | |
| Recombinant DNA reagent | hAAVS1 1 L TALEN | *Sanjana et al., 2012* | Plasmid #35431 | hAAVS1 1 L TALEN was a gift from Feng Zhang (Addgene plasmid # 35431; http://n2t.net/addgene: 35431; RRID:Addgene_35431) |
| Recombinant DNA reagent | hAAVS1 1 R TALEN | *Sanjana et al., 2012* | Plasmid #35432 | hAAVS1 1 R TALEN was a gift from Feng Zhang (Addgene plasmid # 35432; http://n2t.net/addgene: 35432; RRID:Addgene_35432) |
| Recombinant DNA reagent | β-OGT | *Park et al., 2017* | N/A | β-globin based OGT splicing reporter |
| Commercial assay or kit | CellTiter-Glo | Promega | Cat #G7570 | |
| Commercial assay or kit | AMPure XP | Beckman Coulter | Cat #A63880 | |
| Commercial assay or kit | KAPA mRNA HyperPrep Kit | Roche | Cat # KK8580 | |
| Sequence-based reagent | Negative Control No. 2 siRNA | ThermoFisher | Cat #4390846 | Non-target siRNA |
| Sequence-based reagent | SFSWAP siRNA #1 | ThermoFisher | Assay ID #s12746 | Silencer select siRNA |
| Sequence-based reagent | SFSWAP siRNA #2 | ThermoFisher | Assay ID #s12748 | Silencer select siRNA |
| Sequence-based reagent | UPF1 siRNA #1 | Sigma-Aldrich | siRNA ID # SASI_Hs01_00101017 | Mission siRNA |
| Sequence-based reagent | UPF1 siRNA #2 | Sigma-Aldrich | siRNA ID # SASI_Hs01_00101018 | Mission siRNA |
| Sequence-based reagent | OGT siRNA #1 | ThermoFisher | Assay ID #s16094 | Silencer select siRNA |

*Appendix 1 Continued on next page*

*Appendix 1 Continued*

| Reagent type (species) or resource | Designation | Source or reference | Identifiers | Additional information |
|---|---|---|---|---|
| Sequence-based reagent | OGT siRNA #2 | ThermoFisher | Assay ID #s16095 | Silencer select siRNA |
| Sequence-based reagent | NIPP1 siRNA #1 | ThermoFisher | Assay ID #s10954 | Silencer select siRNA |
| Sequence-based reagent | NIPP1 siRNA #2 | ThermoFisher | Assay ID #s10955 | Silencer select siRNA |
| Sequence-based reagent | LSM4 siRNA #1 | ThermoFisher | Assay ID #s24521 | Silencer select siRNA |
| Sequence-based reagent | LSM4 siRNA #2 | ThermoFisher | Assay ID #s24522 | Silencer select siRNA |
| Sequence-based reagent | BCAS2 siRNA #1 | ThermoFisher | Assay ID #s20104 | Silencer select siRNA |
| Sequence-based reagent | BCAS2 siRNA #2 | ThermoFisher | Assay ID #s20105 | Silencer select siRNA |
| Sequence-based reagent | ELAVL1 siRNA #1 | ThermoFisher | Assay ID #s4609 | Silencer select siRNA |
| Sequence-based reagent | ELAVL1 siRNA #2 | ThermoFisher | Assay ID #4610 | Silencer select siRNA |
| Sequence-based reagent | HNRNPU siRNA #1 | ThermoFisher | Assay ID #s6745 | Silencer select siRNA |
| Sequence-based reagent | HNRNPU siRNA #2 | ThermoFisher | Assay ID #s6744 | Silencer select siRNA |
| Sequence-based reagent | ZNF236 siRNA #1 | ThermoFisher | Assay ID #s15328 | Silencer select siRNA |
| Sequence-based reagent | ZNF236 siRNA #2 | ThermoFisher | Assay ID #s15326 | Silencer select siRNA |
| Sequence-based reagent | ZC3H13 siRNA #1 | ThermoFisher | Assay ID #s23011 | Silencer select siRNA |
| Sequence-based reagent | ZC3H13 siRNA #2 | ThermoFisher | Assay ID #s23012 | Silencer select siRNA |
| Sequence-based reagent | ZNHIT2 siRNA #1 | ThermoFisher | Assay ID #s194322 | Silencer select siRNA |
| Sequence-based reagent | ZNHIT2 siRNA #2 | ThermoFisher | Assay ID #s2212 | Silencer select siRNA |
| Sequence-based reagent | SF1 siRNA #1 | ThermoFisher | Assay ID #s14976 | Silencer select siRNA |
| Sequence-based reagent | SF1 siRNA #2 | ThermoFisher | Assay ID #s200464 | Silencer select siRNA |
| Sequence-based reagent | KIAA1429 siRNA #1 | ThermoFisher | Assay ID #s24832 | Silencer select siRNA |
| Sequence-based reagent | KIAA1429 siRNA #2 | ThermoFisher | Assay ID #s24833 | Silencer select siRNA |
| Software, algorithm | FlowJo | BD Biosciences | v 10 | |
| Software, algorithm | Snapgene | Dotmatics | v 7.2.1 | |
| Software, algorithm | rMATS | *Shen et al., 2014* | v 4.3.0 | |
| Software, algorithm | STAR | *Dobin et al., 2013* | v 2.7 | |
| Software, algorithm | Cutadapt | DOI:10.14806/ej.17.1.200 | v 1.9.1 | |

*Appendix 1 Continued on next page*

*Appendix 1 Continued*

| Reagent type (species) or resource | Designation | Source or reference | Identifiers | Additional information |
|---|---|---|---|---|
| Software, algorithm | FastQC | http://www.bioinformatics.babraham.ac.uk/projects/fastqc/ | v 0.11.5 | |
| Software, algorithm | Rstudio | Posit Software | 2024.04.1 | |
| Software, algorithm | MAGeCK-VISPR | *Li et al., 2015* | v 0.5.6 | |
| Software, algorithm | Whippet | *Kreppel et al., 1997*; *Sterne-Weiler et al., 2018* | v 1.6.1 | |
| Software, algorithm | MAJIQ | *Vaquero-Garcia et al., 2023* | v 2.4.dev102 | |
| Software, algorithm | maser | https://github.com/DiogoVeiga/maser | v 1.22.0 | |
| Software, algorithm | bedtools | *Quinlan and Hall, 2010* | v 2.29.0 | |
| Software, algorithm | minimap2 | *Li, 2018* | v 2.26 | |
| Software, algorithm | edgeR | *Robinson et al., 2010* | v 4.2.1 | |
| Software, algorithm | DESeq2 | *Love et al., 2014* | v 1.44.0 | |
| Software, algorithm | EDASeq | *Risso et al., 2011* | v 2.38.0 | |
| Software, algorithm | RUVSeq | *Risso et al., 2014* | v 1.38.0 | |
| Software, algorithm | rMAPS2 web server | *Hwang et al., 2020* | N/A | http://rmaps.cecsresearch.org/ |
| Software, algorithm | rmats2sashimiplot | https://github.com/Xinglab/rmats2sashimiplot | v 3.0.0 | |
| Software, algorithm | Graphpad Prism | Dotmatics | v 10.3.0 | |

