## [Editor Report · eLife Assessment]

This study provides **fundamental** insights into the regulation of a retained intron in the mRNA coding for OGT, a process known to be regulated by the O-GlcNAc cycling system, and highlights the functional role of the splicing regulator SFSWAP. The evidence supporting the claims of the authors is **convincing**; the authors performed an elegant state-of-the-art CRISPR knockout strategy and sophisticated bioinformatic analysis to identify SFSWAP as a negative regulator of alternative splicing. The work will be of interest to researchers in the fields of splicing and glycobiology.

---

## [Referee Report · Reviewer #1 (Public review)]

Summary:

Govindan and Conrad use a genome-wide CRISPR screen to identify genes regulating retention of intron 4 in OGT, leveraging an intron retention reporter system previously described (PMID: 35895270). Their OGT intron 4 reporter reliably responds to O-GlcNAc levels, mirroring the endogenous splicing event. Through a genome-wide CRISPR knockout library, they uncover a range of splicing-related genes, including multiple core spliceosome components, acting as negative regulators of OGT intron 4 retention. They choose to follow up on SFSWAP, a largely understudied splicing regulator shown to undergo rapid phosphorylation in response to O-GlcNAc level changes (PMID: 32329777). RNA-sequencing reveals that SFSWAP depletion not only promotes OGT intron 4 splicing but also broadly induces exon inclusion and intron splicing, affecting decoy exon usage. While this study offers interesting insights into intron retention regulation and O-GlcNAc signaling, the RNA-Sequencing experiments lack essential controls needed to provide full confidence to the authors' conclusions.

Strengths:

(1) This study presents an elegant genetic screening approach to identify regulators of intron retention, uncovering core spliceosome genes as unexpected positive regulators of intron retention.

(2) The work proposes a novel functional role for SFSWAP in splicing regulation, suggesting that it acts as a negative regulator of splicing and cassette exon inclusion, which contrasts with expected SR-related protein functions.

(3) The authors suggest an intriguing model where SFSWAP, along with other spliceosome proteins, promotes intron retention by associating with decoy exons.

Weaknesses:

(1) The conclusions regarding SFSWAP's impact on alternative splicing rely on cells treated with a single pool of two siRNAs for five days. The absence of independent siRNA treatments raises concerns about potential off-target effects, which may reduce confidence in the observed SFSWAP-dependent splicing changes. Rescue experiments or using additional independent siRNA treatments would strengthen the conclusions.

(2) The mechanistic role of SFSWAP in splicing would benefit from further exploration, though this may be more appropriate for future studies.

Comments on revisions:

The authors have addressed all my previous recommendations.

---

## [Referee Report · Reviewer #2 (Public review)]

Summary:

The paper describes an effort to identify the factors responsible for intron retention and alternate exon splicing in a complex system known to be regulated by the O-GlcNAc cycling system. The CRISPR/Cas9 system was used to identify potential factors. The bioinformatic analysis is sophisticated and compelling. The conclusions are of general interest and advance the field significantly.

Strengths:

- Exhaustive analysis of potential splicing factors in an unbiased screen.

- Extensive genome wide bioinformatic analysis.

- Thoughtful discussion and literature survey

Weaknesses:

- No firm evidence linking SFSWA to an O-GlcNAc specific mechanism.

- Resulting model leaves many unanswered questions.

Comments on revisions:

I think the authors have adequately dealt with the overall reviewer's comments.

---

## [Referee Report · Reviewer #3 (Public review)]

Summary:

The major novel finding in this study is that SFSWAP, a splicing factor containing an RS domain but no canonical RNA binding domain, functions as a negative regulator of splicing. More specifically, it promotes retention of specific introns in a wide variety of transcripts including transcripts from the OGT gene previously studied by the Conrad lab. The balance between OGT intron retention and OGT complete splicing is an important regulator of O-GlcNAc expression levels in cells.

Strengths:

An elegant CRISPR knockout screen employed a GFP reporter, in which GFP is efficiently expressed only when the OGT retained intron is removed (so that the transcript will be exported from the nucleus to allow for translation of GFP). Factors whose CRISPR knockdown cause decreased intron retention therefore increase GFP, and these can be identified by sequencing RNA of GFP-sorted cells. SFSWAP was thus convincingly identified as a negative regulator of OGT retained intron splicing. More focused studies of OGT intron retention indicate that it may function by regulating a decoy exon previously identified in the intron, and that this may extend to other transcripts with decoy exons.

Weaknesses:

The mechanism by which SFSWAP represses retained introns is unclear, although some data suggests it can operate (in OGT) at the level of a recently reported decoy exon within that intron. Interesting / appropriate speculation about possible mechanism are provided and will likely be the subject of future studies.

Overall the study is well done and carefully described.

---

## [Author Response]

The following is the authors’ response to the original reviews

**Public Reviews:**

**Reviewer #1 (Public review):**
Summary:Govindan and Conrad use a genome-wide CRISPR screen to identify genes regulating retention of intron 4 in OGT, leveraging an intron retention reporter system previously described (PMID: 35895270). Their OGT intron 4 reporter reliably responds to O-GlcNAc levels, mirroring the endogenous splicing event. Through a genome-wide CRISPR knockout library, they uncover a range of splicing-related genes, including multiple core spliceosome components, acting as negative regulators of OGT intron 4 retention. They choose to follow up on SFSWAP, a largely understudied splicing regulator shown to undergo rapid phosphorylation in response to O-GlcNAc level changes (PMID: 32329777). RNA-sequencing reveals that SFSWAP depletion not only promotes OGT intron 4 splicing but also broadly induces exon inclusion and intron splicing, affecting decoy exon usage. While this study offers interesting insights into intron retention and O-GlcNAc signaling regulation, the RNA sequencing experiments lack the essential controls needed to provide full confidence to the authors' conclusions.Strengths:(1) This study presents an elegant genetic screening approach to identify regulators of intron retention, uncovering core spliceosome genes as unexpected positive regulators of intron retention.(2) The work proposes a novel functional role for SFSWAP in splicing regulation, suggesting that it acts as a negative regulator of splicing and cassette exon inclusion, which contrasts with expected SR-related protein functions.(3) The authors suggest an intriguing model where SFSWAP, along with other spliceosome proteins, promotes intron retention by associating with decoy exons.

We thank the reviewer for recognizing and detailing the strengths of our manuscript.

Weaknesses:(1) The conclusions on SFSWAP impact on alternative splicing are based on cells treated with two pooled siRNAs for five days. This extended incubation time without independent siRNA treatments raises concerns about off-target effects and indirect effects from secondary gene expression changes, potentially limiting confidence in direct SFSWAP-dependent splicing regulation. Rescue experiments and shorter siRNA-treatment incubation times could address these issues.

We repeated our SFSWAP knockdown analysis and analyzed both OGT e4-e5 junction splicing and SFSWAP transcript levels by RT-qPCR (now included in Sup. Fig. S4) from day 2 to day 5 post siRNA treatment. We observed that the time point at which OGT intron 4 removal increases (day 2) coincides with the time at which SFSWAP transcript levels start decrease, consistent with a direct effect of SFSWAP knockdown on OGT intron 4 splicing. Moreover, the effect of SFSWAP knockdown on OGT intron 4 splicing peaks between day 4-5, supporting our use of these longer time points to cast a wide net for SFSWAP targets.

(2) The mechanistic role of SFSWAP in splicing would benefit from further exploration. Key questions remain, such as whether SFSWAP directly binds RNA, specifically the introns and exons (including the decoy exons) it appears to regulate. Furthermore, given that SFSWAP phosphorylation is influenced by changes in O-GlcNAc signaling, it would be interesting to investigate this relationship further. While generating specific phosphomutants may not yield definitive insights due to redundancy and also beyond the scope of the study, the authors could examine whether distinct SFSWAP domains, such as the SR and SURP domains, which likely overlap with phosphorylation sites, are necessary for regulating OGT intron 4 splicing.

We absolutely agree with the reviewer that the current work stops short of a detailed mechanistic study, and we have made every attempt to be circumspect in our interpretations to reflect that limitation. In addition, we are very interested in delving more deeply into the mechanistic aspects of this regulation. In fact, we have initiated many of the experiments suggested by the reviewer (and more), but in each case, rigorous interpretable results will require a minimum another year’s time.

For example, we have used crosslinking and biotin labeling techniques (using previously available reagents from Eclipsebio) to test whether SFSWAP binds RNA. The results were negative, but the lack of strong SFSWAP antibodies required that we use a transiently expressed myc-tagged SFSWAP. Therefore, this negative result could be an artifact of the exogenous expression and/or tagging. Given the difficulties of “proving the negative”, considerably more work will be required to substantiate this finding. As another example, we intend to develop a complementation assay as suggested. For an essential gene, the ideal complementation system employs a degron system, and we have spent months attempting to generate a homozygous AID-tagged SFSWAP. Unfortunately, we so far have only found heterozygotes. Of course, this could be because the tag interferes with function, the insert was not efficiently incorporated by homologous repair, or that we simply haven’t yet screened a sufficient number of clones. We’re confident that these technical issues that can be addressed, but they will take a significant amount of time to resolve. While we would ideally define a mechanism, we think that the data reported here outlining functions for SFSWAP in splicing represent a body of work sufficient for publication.

(3) Data presentation could be improved (specific suggestions are included in the recommendations section). Furthermore, Excel tables with gene expression and splicing analysis results should be provided as supplementary datasheets. Finally, a more detailed explanation of statistical analyses is necessary in certain sections.

We have addressed all specific suggestions as detailed in the recommendations below.

**Reviewer #2 (Public review):**
Summary:The paper describes an effort to identify the factors responsible for intron retention and alternate exon splicing in a complex system known to be regulated by the O-GlcNAc cycling system. The CRISPR/Cas9 system was used to identify potential factors. The bioinformatic analysis is sophisticated and compelling. The conclusions are of general interest and advance the field significantly.Strengths:(1) Exhaustive analysis of potential splicing factors in an unbiased screen.(2) Extensive genome wide bioinformatic analysis.(3) Thoughtful discussion and literature survey.

We thank the reviewer for recognizing and detailing the strengths of our manuscript.

Weaknesses:(1) No firm evidence linking SFSWAP to an O-GlcNAc specific mechanism.

We couldn’t agree more with this critique. Indeed, our intention at the outset for the screen was to find an O-GlcNAc sensor linking OGT splicing with O-GlcNAc levels. As often occurs with high-throughput screens, we didn’t find exactly what we were looking for, but the screen nonetheless pointed us to interesting biology. Prompted by our screen, we describe new insights into the function of SFSWAP a relatively uncharacterized essential gene. Currently, we are testing other candidates from our screen, and we are performing additional studies to identify potential O-GlcNAc sensors.

(2) Resulting model leaves many unanswered questions.

We agree (see Reviewer 1, point 2 response).

**Reviewer #3 (Public review):**
Summary:The major novel finding in this study is that SFSWAP, a splicing factor containing an RS domain but no canonical RNA binding domain, functions as a negative regulator of splicing. More specifically, it promotes retention of specific introns in a wide variety of transcripts including transcripts from the OGT gene previously studied by the Conrad lab. The balance between OGT intron retention and OGT complete splicing is an important regulator of O-GlcNAc expression levels in cells.Strengths:An elegant CRISPR knockout screen employed a GFP reporter, in which GFP is efficiently expressed only when the OGT retained intron is removed (so that the transcript will be exported from the nucleus to allow for translation of GFP). Factors whose CRISPR knockdown causes decreased intron retention therefore increase GFP, and can be identified by sequencing RNA of GFP-sorted cells. SFSWAP was thus convincingly identified as a negative regulator of OGT retained intron splicing. More focused studies of OGT intron retention indicate that it may function by regulating a decoy exon previously identified in the intron, and that this may extend to other transcripts with decoy exons.

We thank the reviewer for recognizing the strengths of our manuscript.

Weaknesses:The mechanism by which SFSWAP represses retained introns is unclear, although some data suggests it can operate (in OGT) at the level of a recently reported decoy exon within that intron.

Interesting/appropriate speculation about possible mechanisms are provided and will likely be the subject of future studies.

We completely agree that this is a limitation of the current study (see above). Now that we have a better understanding of SFSWAP functions, we will continue to explore SFSWAP mechanisms as suggested.

Overall the study is well done and carefully described but some figures and some experiments should be described in more detail.
**Recommendations for the authors:**

**Reviewer #1 (Recommendations for the authors):**
(1) Clarify and add missing statistical details across the figures. For example, Figure S2 lacks statistical comparisons, and in Figures 4A and 4C the tests applied should be specified in the legend.

We have added appropriate statistical analysis wherever missing and edited figure legends to specify the tests used.

(2) The authors are strongly encouraged to provide detailed tables of gene expression and alternative splicing analyses from RNA-Seq experiments (e.g., edgeR, rMATS, Whippet, and MAJIQ), as this would enhance transparency and facilitate data interpretation.

We have added tables for gene expression and alternate splicing analysis as suggested Suppl. tables 3-

6).

(3) Although the legend sometimes indicates differently (e.g., Figure 3b, 5a, 5c, etc), the volcano plots showing the splicing changes do not contain a cutoff for marginally differential percent spliced in or intron retention values.

The legends have been edited to reflect the correct statistical and/or PSI cutoffs.

(4) For consistency, use a consistent volcano plot format across all relevant figures (Figures 3b, 5a-c, S3, S4, S7, and S8), including cutoffs for differential splicing and the total count of up- and down-regulated events.

Due to different statistical frameworks and calculations employed by different alternate splicing pipelines, we could not use the same cutoffs for different pipelines. However, we have now indicated the number of up- and down-regulated events for consistency among the volcano plots.

(5) What is the overlap of differentially regulated events between the different analytical methodologies applied?

We analyzed the degree of overlap between the three pipelines used in the paper using a Venn diagram (added to Suppl. Fig. S7). However, as widely reported in literature (e.g., Olofsson et al., 2023; Biochem Biophys Res Commun. 2023; doi: 10.1016/j.bbrc.2023.02.053.), the degree of overlap between pipelines is quite low.

(6) To further substantiate your conclusions, additional validations of RNA-Seq splicing data, ideally visualized on an agarose gel, would be valuable, especially for exons and introns regulated by SFSWAP, and particularly for OGT decoy exons in Figure 4c.

We have not included these experiments as we focused on other critiques for this resubmission. Because the RNA-seq, RT-PCR and RT-qPCR data all align, we are confident that the products we are seeing are correctly identified and orthogonally validated (Figs 2d, 4a, 4b, and 4c).

(7) It would be more informative if the CRISPR screen data were presented in a format where both the adjusted p-value and LFC values of the hits are presented. Perhaps a volcano plot?

We have now included these graphs in revised Supplementary Figure S2.

(8) In Figure 2d, a cartoon showing primer binding sites for each panel could aid interpretation, particularly in explaining the unexpected simultaneous increase in OGT mRNA and intron retention upon SFSWAP knockdown.

We have added a cartoon showing primer binding sites similar to that shown in Fig. 4a.

(9) Page 9, line 1, states that SFSWAP autoregulates its expression by controlling intron retention. Including a Sashimi plot would provide visual support for this claim.

The data suggesting that SFSWAP autoregulates its own transcript abundance were reported in Zachar et al. (1994), not from our own studies. Validation of those data with our RNA-seq data is confounded by the fact that we are using siRNAs to knockdown the SFSWAP RNA at the transcript level (Fig. S15).

(10) In the legend of Figure S2 the authors state that negative results are inconclusive because RNA knockdowns are not verified by western blotting or qRT-PCR. This is correct, but the reviewer would also argue that the positive results are also inconclusive as they are not supported by a rescue experiment to confirm that the effect is not due to off-target effects.

This is a fair point with respect to the siRNA experiments on their own. However, the CRISPR screen was performed with sgRNAs, and MAGeCK RRA scores are high only for those genes that have multiple sgRNAs that up-regulate the gene. Examination of the SFSWAP sgRNAs individually shows that three of four SFSWAP sgRNAs had false discovery rates ≤10^-42^ for GFP upregulation. Thus, the siRNAs provide an additional orthogonal approach. It seems unlikely that the siRNAs, and three independent sgRNAs will have the same off-target results. Thus, these combined observations support the conclusion that SFSWAP loss leads to decreased OGT intron retention.

(11) For clarity in Figure 3a, consider using differential % spliced in or intron retention bar plots with directionality (positive and negative axis) and labeling siSFSWAP as the primary condition.(12) Consider presenting Figure 5D as a box plot with a Wilcoxon test for statistical comparison.

For both points 11 and 12, we have tried the graphs as the reviewer suggested. While these were good suggestions, in both cases we felt that the original plots ended up presenting a clearer presentation of the data (see Author response image 1).

**Author response image 1. sa4fig1:** 

(13) Please expand the Methods section to detail the Whippet and MAJIQ analyses.

We have expanded the methods section to include additional details of the alternate splicing analysis.

(14) Include coordinates for the four possible OGT decoy exon combinations analyzed in the Methods section.

We have added the coordinates of all four decoy forms in the methods section.

(15) A section on SFSWAP mass spectrometry is listed in Methods but is missing from the manuscript.

This section has now been removed.

**Reviewer #2 (Recommendations for the authors):**
This is an excellent contribution. The paper describes an effort to identify the factors responsible for intron retention and alternate exon splicing in a complex system known to be regulated by the O-GlcNAc cycling system. The CRISPR/Cas9 system was used to identify potential factors. The bioinformatic analysis is sophisticated and compelling. The conclusions are of general interest and advance the field significantly.Some specific recommendations.(1) The plots in Figure 3 describing SI and ES events are confusing to this reader. Perhaps the violin plot is not the best way to visualize these events. The same holds true for the histograms in the lower panel of Figure 3. Not sure what to make of these plots.

For Figure 3b, we include both scatter and violin plots to represent the same data in two distinct ways. For Figure 3d, we agree that these are not the simplest plots to understand, and we have spent significant time trying to come up with a better way of displaying these trends in GC content as they relate to SE and RI events. Unfortunately, we were unable to identify a clearer way to present these data.

(2) The model (Figure 6) is very useful but confusing. The legend and the Figure itself are somewhat inconsistent. The bottom line of the figure is apparent but I fear that the authors are trying to convey a more complete model than is apparent from this figure. Please revise.

We have simplified the figure from the previous submission. As mentioned above, we admit that mechanistic details remain unknown. However, we have tried to generate a model that reflects our data, adds some speculative elements to be tested in the future, but remains as simple as possible. We are not quite sure what the reviewer was referring to as “somewhat inconsistent”, but we have attempted to clarify the model in the revised Discussion and Figure legend.

(3) It is unclear how normalization of the RNA seq experiments was performed (eg. Figure S5 and 6).

The normalization differences in Fig. S5 and S6 (now Fig S8 and S9) were due to scaling differences during the use of rmats2sashimiplot software. We have now replaced Fig. S5 to reflect correctly scaled images.

I am enthusiastic about the manuscript and feel that with some clarification it will be an important contribution.

Thank you for these positive comments about our study!

**Reviewer #3 (Recommendations for the authors):**
(1) In Figure 1f, it is clear that siRNA-mediated knockdown of OGT greatly increases spliced RNA as the cells attempt to compensate by more efficient intron removal (three left lanes). However, there is no discussion of the various treatments with TG or OSMI. Might quantitation of these lanes not also show the desired effects of TG and OSMI on spliced transcript levels?

The strong effect of OGT knockdown masks the (comparatively modest) effects of subsequent inhibitor treatments on the reporter RNA. We have edited the results section to clarify this.

(2) In Figure 2c, why is the size difference between spliced RNA and intron-retained RNA so different in the GFP-probed gel (right) compared with the OGT-probed gel (left)? Even recognizing that the GFP probe is directed against reporter transcripts, and the OGT probe (I think) is directed against endogenous OGT transcripts, shouldn't the difference between spliced and unspliced bands be the same, i.e., +/- the intron 4 sequence. Also, why does the GFP probe detect the unspliced transcript so poorly?

The fully spliced endogenous OGT mRNA is ~5.5 kb while the fully spliced reporter is only ~1.6kb, so the difference in size (the apparent shift relative to the mRNA) is quite different. Moreover, the two panels in Fig 2c are not precisely scaled to one another, so direct comparisons cannot be made.

The intron retained isoform does not accumulate to high levels in this reporter, a phenotype that we also observed with our GFP reporter designed to probe the regulation of the MAT2A retained intron (Scarborough et al., 2021). We are not certain about the reason for these observations, but suspect that the reporter RNA’s retained intron isoforms are less stable in the nucleus than their endogenous counterparts. Alternatively, the lack of splicing may affect 3´ processing of the transcripts so that they do not accumulate to the high levels observed for the wild-type genes.

(3) Please provide more information about the RNA-seq experiments. How many replicates were performed under each of the various conditions? The methods section says three replicates were performed for the UPF1/TG experiments; was this also true for the SFSWAP experiments?

All RNA-seq experiments were performed in biological triplicates. We have edited the methods section to clarify this.

(4) Relatedly, the several IGV screenshots shown in Figure 3C presumably represent the triplicate RNA seq experiments. In part D, how many experiments does the data represent? Is it a compilation of three experiments?

Fig. 3d is derived from alternate splicing analysis performed on three biological replicates. We have added the number of replicates (n=3) on the figure to clarify this. We have also noted that the three IGV tracks represent biological replicates in the Figure legend for 3c.

(5) Please provide more details regarding the qRT-PCR experiments.

We have provided the positions of primer sets used for RT-qPCR analysis and cartoon depictions of target sites below the data wherever appropriate.

(6) In the discussion of decoy exon function (in the Discussion section), several relevant observations are cited to support a model in which decoy exons promote assembly of splicing factors. One might also cite the finding that eCLIP profiling has found enriched binding of U2AF1 and U2AF2 at the 5' splice site region of decoy exons (reference 16).

Excellent point. This has now been added to the Discussion.

Minor corrections / clarifications:(1) In the Figure 2A legend, CRISPR is misspelled.

Corrected.

(2) In the discussion, the phrase "indirectly inhibits splicing of exons 4 and 5, but promoting stable unproductive assembly of the spliceosome", the word "but" should probably be "by".

Corrected.